# CHASE-SQL: Multi-Path Reasoning and Preference Optimized Candidate Selection in Text-to-SQL

**Mohammadreza Pourreza**[1][*], **Hailong Li**[1][*], **Ruoxi Sun**[1], **Yeounoh Chung**[1], **Shayan Talaei**[2],
**Gaurav Tarlok Kakkar**[1], **Yu Gan**[1], **Amin Saberi**[2], **Fatma Özcan**[1], **Sercan Ö. Arık**[1]
[1]Google Cloud, Sunnyvale, CA, USA
[2]Stanford University, Stanford, CA, USA
{pourreza, hailongli, ruoxis, yeounoh}@google.com
{gkakkar, gany, fozcan, soarik}@google.com
{stalaei, saberi}@stanford.edu
[*]Equal contribution

## Abstract

We present CHASE-SQL, a novel framework addressing large language model (LLM) performance challenges for Text-to-SQL tasks by leveraging multi-agent modeling and test-time compute for improved candidate generation and selection. CHASE-SQL uses LLMs to generate diverse SQL candidates with: (1) a divide-and-conquer approach to break down complex queries, (2) chain-of-thought reasoning based on query execution plans, and (3) instance-aware synthetic example generation for tailored few-shot demonstrations. A selection agent ranks candidates via pairwise comparisons using a fine-tuned binary selection LLM, offering robust performance. This framework improves SQL query quality and diversity, achieving state-of-the-art execution accuracy of 73.0% on the BIRD Text-to-SQL benchmark test set, topping the leaderboard at the time of submission.

## 1 Introduction

Text-to-SQL, as a bridge between human language and machine-readable structured query languages, is crucial for many use cases, converting natural language questions into executable SQL commands (Androutsopoulos et al., 1995; Hristidis et al., 2003; Li & Jagadish, 2014; Li et al., 2024c; Yu et al., 2018). By enabling users to interact with complex database systems without requiring SQL proficiency, Text-to-SQL empowers users to extract valuable insights, perform streamlined data exploration, make informed decisions, generate data-driven reports and mine better features for machine learning (Wang et al., 2019; Pourreza & Rafiei, 2024a; Sun et al., 2023; Chen et al., 2023; Pourreza et al., 2024; Pérez-Mercado et al., 2023; Xie et al., 2023). Furthermore, Text-to-SQL systems play a pivotal role in automating data analytics with complex reasoning and powering conversational agents, expanding their applications beyond traditional data retrieval (Sun et al., 2023; Xie et al., 2023). As data continues to grow exponentially, the ability to query databases efficiently without extensive SQL knowledge becomes increasingly vital for a broad range of applications.

Text-to-SQL can be considered a specialized form of code generation, with the contextual information potentially including the database schema, its metadata and along with the values. In the broader code generation domain, utilizing LLMs to generate a wide range of diverse candidates and select the best one has proven to be effective (Li et al., 2022; Ni et al., 2023; Chen et al., 2021). However, it is non-obvious what leads to most effective candidate proposal and winner selector mechanisms. A straightforward yet effective approach involves generating candidates using zero-/few-shot or open-ended prompting, followed by selecting the best options utilizing self-consistency (Wang et al., 2022), which entails clustering candidates based on their execution outputs. This approach has demonstrated promising results in several studies (Maamari et al., 2024; Talaei et al., 2024; Lee et al., 2024; Wang et al., 2023). However, a single prompt design might not fully unleash the extensive Text-to-SQL knowledge of LLMs, and self-consistency methods might not be always effective.

In fact, as illustrated in Table 1, the most consistent answers would not always be the correct ones, with an upper-bound performance 14% higher than that achieved through self-consistency. This substantial gap highlights the potential for significant improvement by implementing more effective selection methods to identify the best answer from the pool of candidate queries.

Building on the challenges outlined in the previous section, we propose novel approaches to improve LLM performance for Text-to-SQL by leveraging judiciously-designed test-time computations in an agentic framework. As indicated by the upper bound in Table 1, utilizing LLMs' intrinsic knowledge offers significant potential for improvement. We propose methods that generate a diverse set of high-quality candidate responses and apply a selection mechanism to identify the best answer. Achieving both high-quality and diverse candidate responses is critical for the success of scoring-based selection methods. Low diversity limits improvement potential and reduces the difference between self-consistency and scoring-based

Table 1: Evaluating single-query generation vs. ensemble methods of self-consistency and the upper bound that can be achieved for Text-to-SQL with Gemini 1.5 Pro on the BIRD dev set. EX stands for execution accuracy.

| Method | EX (%) |
| --- | --- |
| Single query | 63.01 |
| Self-consistency | 68.84 (+ 5.84) |
| Upper-bound | **82.79** (+ 19.78) |

approaches. While techniques like increasing temperature or reordering prompt contents can boost diversity, they often compromise the quality of the candidates. To address this, we introduce effective candidate generators designed to enhance diversity while maintaining high-quality outputs. Specifically, we propose three distinct candidate generation approaches, each capable of producing high-quality responses. The first is inspired by the divide-and-conquer algorithm, which breaks down complex problems into smaller, manageable parts to handle difficult queries. The second employs a query execution-plan-based chain-of-thought strategy, where the reasoning process mirrors the steps a database engine takes during query execution. Lastly, we introduce a novel online synthetic example generation method, which helps the model better understand the underlying data schema of the test database. These methods, when used independently, can produce highly-accurate SQL outputs. To effectively select the best answer among candidates, we introduce a selection agent, trained with a classification objective, that assigns scores based on pairwise comparisons between candidate queries. With this agent, we construct a comparison matrix for all candidates and select the final response based on the highest cumulative score. By combining these candidate generation methods with the proposed scoring model, we create an ensemble approach that leverages the strengths of each strategy to significantly improve overall performance.

We present comprehensive evaluations on the efficacy of proposed methodologies of CHASE-SQL. Our innovative candidate generation approaches demonstrate superior performance compared to traditional generic CoT prompts, illustrating their capability in guiding LLMs through the decomposition of complex problems into manageable intermediate steps. Furthermore, the proposed selection agent significantly outperforms conventional consistency-based methods, contributing to the state-of-the-art results. Specifically, CHASE-SQL reaches an execution accuracy of **73.01%** and **73.0%** on the development set and test set of the challenging BIRD Text-to-SQL dataset which outperforms all of the published and undisclosed methods on this benchmark, by a large margin. Moreover, by leveraging entirely open-source models—Mistral Large Model (AI, 2024) as the candidate generator and a fine-tuned Qwen-2.5-coder model (Team, 2024) as the selector—our method achieved a state-of-the-art performance of 70.33 on the BIRD development set with open-source models.

## 2 METHODS

### 2.1 OVERALL FRAMEWORK

This section outlines the proposed CHASE-SQL framework, which consists of four primary components: 1) Value retrieval, 2) Candidate generator, 3) Query fixer, and 4) Selection agent. As illustrated in Fig. 1. The proposed framework begins by retrieving relevant database values. Subsequently, all contextual information, including retrieved values, database metadata, and schema, is provided to an LLM to generate candidate queries. These candidate queries then undergo a fixing loop, and finally, all candidates are compared in a pairwise way using the trained selection agent to pick the correct answer. The following sections delve into the details of each component.

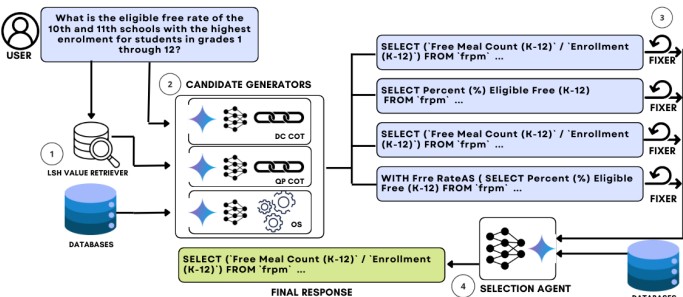

Figure 1: Overview of the proposed CHASE-SQL framework for Text-to-SQL, with value retrieval and using a selection agent for improve picking of the answers among the generated candidates along with a fixer to provide feedback for refinement of the outputs.

## 2.2 VALUE RETRIEVAL

Databases might contain very high number of rows, with often only a few being relevant to a query. Retrieving relevant values is crucial as they can be used in various SQL clauses like 'WHERE' and 'HAVING'. Similar to the approach in (Talaei et al., 2024), we begin by extracting keywords from the given question using an LLM prompted with few-shot examples. For each keyword, we employ locality-sensitive hashing (LSH) (Datar et al., 2004) to retrieve the most syntactically-similar words, and re-rank them based on embedding-based similarity and edit distance. This approach is robust to typos in the question and considers keyword semantics during retrieval.

## 2.3 MULTI-PATH CANDIDATE GENERATION

As shown in Table 1, relying solely on consistency among responses can lead to sub-optimal performance. Therefore, we prioritize diversity in generation of multiple response candidates to increase the likelihood of generating at least one correct answer. Among the diverse responses generated by the candidate generators, we select one as the final response using a selection agent that compares candidates pairwise. To generate diverse responses, we increase the next token sampling temperature, and also shuffle the order of columns and tables in the prompt.

Chain-of-Thought (CoT) prompting (Wei et al., 2022) has been proposed to enhance LLMs' reasoning abilities by conditioning their final responses on a step-by-step chain of reasoning. Most CoT prompting approaches rely on few-shot examples in the prompt to guide LLMs on thinking step-by-step, following the format $M = (q_i, r_i, s_i)$, where $q_i$ is the example question, $r_i$ is the reasoning path, and $s_i$ is the ground truth SQL query for $q_i$. We employ two distinct reasoning methods and an online synthetic example generation approach. As shown in Fig. 3a, different generators can yield different outputs, indicating their effectiveness for specific questions and databases.

**Divide and Conquer CoT:** Divide-and-conquer perspective brings breaking down complex problems into smaller sub-problems, solving each individually, and then combining the solutions to obtain the final answer. Along these lines, we propose a CoT prompting approach that first decomposes the given question into smaller sub-problems using pseudo-SQL queries. In the 'conquer' step, the solutions to these sub-problems are aggregated to construct the final answer. Finally, an optimization step is applied to the constructed query to remove redundant clauses and conditions. This approach is particularly powerful handling complex scenarios that involve nested queries, e.g. intricate WHERE or HAVING conditions, and queries requiring advanced mathematical operations. In Appendix Fig. 18, we exemplify a question and its corresponding SQL query that was successfully solved using this generator, a scenario the other methods considered in this paper could not address due to the query's complex conditions and SQL clauses. For a more detailed view of the divide-and-conquer prompt, please see Appendix Fig. 17. Additionally, Alg. 1 outlines the step-by-step process of this strategy to generate the final SQL output using a single LLM call.

**Query Plan CoT:** A query (execution) plan is a sequence of steps that the database engine follows to access or modify the data described by a SQL command. When a SQL query is executed, the database management systems' query optimizers translate the SQL text into a query plan that the database engine can execute. This plan outlines how tables are accessed, how they are joined, and

---

**Algorithm 1** Divide and Conquer Chain-of-Thought (CoT) Strategy for Text-to-SQL.

---

**Require:** Set of human-annotated few-shot examples $M$, user question $Q_u$, target database $D$ associated
    with the question, and a large language model (LLM) $\theta$.
    **Divide:**
1:  $S_q \leftarrow \theta(M, D, Q_u)$ *// Decompose the original question $Q_u$ into a set of sub-questions $S_q$*
2:  $S_{sql} \leftarrow \emptyset$ *// Initialize an empty set $S_{sql}$ to store partial SQL queries for each sub-question*
    **Conquer:**
3:  **for** each sub-question $q_i$ in $S_q$ **do**
4:      *// Generate a partial SQL query for each sub-question $q_i$*
5:      $S_{sql} \leftarrow S_{sql} \cup \{\theta(M, D, Q_u, q_1, ..., q_i, sql_1, ..., sql_{i-1})\}$
6:  **end for**
    **Assemble:**
7:  $S_f \leftarrow \theta(M, D, Q_u, S_q, S_{sql})$ *// Assemble the final SQL query $S_f$ from all sub-queries in $S_{sql}$*
8:  **return** $S_f$

---

the specific operations performed on the data (see Appendix Fig. 20 as an example). Inspired by the step-by-step process that database engines use to execute SQL queries, we propose a reasoning strategy to construct the final SQL output. Query plans for any given SQL query can be obtained using the "EXPLAIN" command, which provides a detailed breakdown of execution steps. However, this output is often presented in a format that is difficult to interpret by LLMs (e.g. in SQLite). To address this, we convert the output of "EXPLAIN" command into a human-readable text format that aligns more closely with the pretraining data of LLMs. The human-readable version of query plans consists of three key steps: (1) identifying and locating the relevant tables for the question, (2) performing operations such as counting, filtering, or matching between tables, and (3) delivering the final result by selecting the appropriate columns to return. This reasoning method complements the divide-and-conquer CoT strategy. While the divide-and-conquer approach is better suited for decomposing complex questions, the query plan approach excels when questions require more reasoning over the relationships between different parts of the question and the database schema. It systematically explains which tables to scan, how to match columns, and how to apply filters. Appendix Fig. 21 shows an example of a question that was answered correctly only by this method. Appendix Fig. 19 provides the prompt used for this reasoning strategy.

**Online Synthetic Example Generation:** Using a set of human-annotated demonstrations for few-shot in-context learning has shown promising results on various related tasks (Pourreza & Rafiei, 2024a). Besides using a few select demonstrations helping with specifying the task and illustrate the step-by-step process deriving the output, question and SQL example pairs are also used for few-shot in-context learning for text-to-SQL (Liu et al., 2022; Nan et al., 2023). While prior works focused on selecting a few handful of relevant examples from existing example pools (e.g., training dataset), we synthesize many example pairs using different schema elements and SQL features per incoming question. Unlike prior few-shot in-context learning approaches, we generate many more than just a few (3-5) examples (Pourreza & Rafiei, 2024a; Li et al., 2024b), as we observe that many-shot learning consistently outperforms few-shot learning (Agarwal et al., 2024).

---

**Algorithm 2** Online Synthetic example generation strategy for Text-to-SQL.

---

**Require:** User question $Q_u$, additional user hint $H_u$, target database $D$ and filtered relevant table columns
    $t$ associated with the question, LLM $\theta$, guidelines $R_f$ for generating examples by SQL features, guide-
    lines $R_t$ for generating examples with filtered schema, and the numbers of examples to generate $n_f$, $n_t$
    respectively
1:  $P \leftarrow \emptyset$ *// $\{(q_i, s_i) \mid q_i, s_i \in \Sigma^*\}$, where $q_i$ is input question , $s_i$ is output SQL for the i-th example*
2:  $P \leftarrow P \cup \{\theta(D, R_f, n_f)\}$ *// Generate n examples with entire database by common SQL features*
3:  $P \leftarrow P \cup \{\theta(t, R_t, n_t)\}$ *// Generate examples with filtered columns to highlight correct schema usage*
4:  **return** $P$

---

Algorithm 2 outlines the online synthetic example generation approach with two LLM generation steps. The first step focuses on generating illustrative examples with common SQL features described in the guideline $R_f$. The SQL features include equality and non-equality predicates, single table and multi-table JOIN, nested JOIN, ORDER BY and LIMIT, GROUP BY and HAVING, various aggregation functions. These are widely applicable SQL clauses and functions – the generated example SQL queries, incorporating these features, follow the BIRD SQL feature distribution (Appendix Fig 25a). The second step focuses on generating examples highlighting correct interpretation of the underlying data schema – the model $\theta$ is asked to generate examples using $t_i$ (column selec-

tion using an approach similar to (Talaei et al., 2024)) and that are similar to the examples outlined in $R_t$. Appendix A.13 provides the prompts used for the example generation).

While a relevant example (e.g. showing a nested JOIN query with multiple tables) can be helpful for questions that require complex JOIN queries, it might also mislead the LLM for overuse (e.g. when a simple single table query is sufficient). This and the inherent ambiguity of natural language query $q_i$, for which we draw the examples by relevance, make the example selection challenging. Thus, we generate and inject the examples to the prompt online per $q_i$. We ask the LLM to generate many input-output pairs for in-context learning. The final set of synthetic examples for $q_i$ contains examples generated with both $R_f$ and $R_t$. This ensures that the example set is diverse both in SQL features/clauses and the choice of relevant tables/columns used. The diversity of the example set is desirable to avoid over-fitting the output to certain patterns (e.g., the model always writes a SQL with JOIN if shown mostly JOIN examples). Mixing various examples for various SQL features and database tables with and without column filtering is observed to result in better generation quality overall (see Appendix Table 8). The generated synthetic examples can guide the model for more accurate text-to-SQL generation. In Appendix A.13, Table 9, we discuss how our synthetic example generation performs compared to selecting examples from often limited examples pools (e.g., training dataset and cross-domain data augmentation (Li et al., 2024b)).

## 2.4 QUERY FIXER

In some cases, LLMs might generate queries that are syntactically incorrect. These queries are clear candidates for correction, as they fail to provide the correct answers. To address this, we apply an LLM-based query fixer that leverages the self-reflection (Shinn et al., 2024) method. The fixer reflects on the previously generated query, using feedback such as syntax error details or empty result sets to guide the correction process. We continue this iterative fixing approach up to a specified number of attempts, $\beta$ (set to three in this paper). Appendix Fig. 22 demonstrates the prompt used for this query fixing step. Additionally, Appendix section A.4 provides a detailed algorithm of the query fixing step.

## 2.5 SELECTION AGENT

With three different methods for generating SQL queries, we can generate a set of candidate queries for any given question. The key challenge in this step is selecting the correct SQL query from this pool of candidates. A naive approach would be to measure consistency among the candidates by executing them, grouping them based on their execution results, and selecting a query from the largest group as the most likely correct answer. However, this would assume that the most consistent answer is always the best one, which is not always the case. Instead, we propose a more refined picking strategy, Algorithm 3, that relies on a selection agent. Given a set of candidates SQL queries $C = \{c_1, c_2, ..., c_n\}$, the final responses are selected by finding the candidate that has the highest score assigned by the selection model. This model $\theta_p$ can take $k$ candidates and rank them based on how accurately each of them answers the given question. Concretely, we formulate the selection of the final response as:

$$c_f = \arg \max_{c \in C} \left( \sum_{i=1}^{\binom{n}{k}} \theta_p(c_{i_1}, \ldots, c_{i_k} \mid Q_u, H_u, D) \right),$$ (1)

where $Q_u$ refers to the user's question, $H_u$ is the provided hint, and $D$ is the target database from which the question is being asked. In Eq. 1, we pass $k$ candidates to the selection model to be ranked, with $k$ being between 1 and $n$. In the extreme case of $k = 1$, the model is unable to make comparisons between candidates, which complicates the evaluation process for the model. As $k$ increases, comparing more candidates makes the process more challenging for the model, as it needs to consider different aspects simultaneously. Consequently, we set $k = 2$ and train a model with a classification objective to compare only two candidates at a time.

Having a set of high-quality and diverse candidates, the most straightforward solution is to employ off-the-shelf LLMs to make pairwise selections. However, experiments with Gemini-1.5-pro showed that using the LLM without fine-tuning resulted in only 58.01% binary classification accuracy. This is primarily due to the candidates being very similar to one another, requiring a fine-tuned

model to learn the nuances and make more accurate decisions. To train the selection agent, we first generate candidate SQL queries on the training set (of Text-to-SQL benchmarks), and group them into clusters based on their execution results. For cases where at least one cluster contains correct queries and others contains incorrect ones, we create training examples in the form of tuples $(Q_u, C_i, C_j, D_{ij}, y_{ij})$, where $Q_u$ is the user's question, $C_i$ and $C_j$ are the two candidate queries being compared, $D_{ij}$ is the database schema used by both candidates (Using the union of candidate schemas is important as it reduces cost and eliminates unnecessary information during comparison.), and $y_{ij} \in 0, 1$ is the label indicating whether $C_i$ or $C_j$ is the correct query. To avoid order bias during training, we randomly shuffle the order of correct and incorrect queries in each pair. Since the number of cases with both correct and incorrect candidates is limited, for instances where no correct candidate exists, we include the ground truth SQL query in the prompt as a hint to guide the model in generating correct candidates.

---

**Algorithm 3** Picking the final SQL query from a pool of candidates.

---

**Require:** Set of candidate SQL queries $C = \{c_1, c_2, ..., c_n\}$, user question $Q_u$, hint $H_u$, target database $D$,
    and a selection model $\theta_p$, $er(c_i, D)$ as the execution result of $c_i$ on $D$
1:  $r_i \leftarrow 0$ for all $c_i \in C$ *// Initialize the score $r_i$ for each candidate query to zero*
2:  **for** each distinct pair $(c_i, c_j)$ where $i \neq j$ **do**
3:      **if** $er(c_i, D) = er(c_j, D)$ **then**
4:         $w \leftarrow i$ *// $c_i$ is the winner if the execution results match*
5:      **else**
6:         $S_{i,j} \leftarrow schema\_union(c_i, c_j, D)$ *// Construct union of schemas used in $c_i$ and $c_j$*
7:         $w \leftarrow \theta_p(S_{i,j}, Q_u, H_u, c_i, c_j) w \in \{i, j\}$ *// Use binary classifier $\theta_p$ to select the winner, $w \in \{i, j\}$*

8:      **end if**
9:      $r_w \leftarrow r_w + 1$ *// Increase the score of the winner $c_w$ by 1*
10: **end for**
11: $c_f \leftarrow \arg\max_{c_i \in C} r_i$ *// Select the candidate with the highest score as the final SQL query $c_f$*
12: **return** $c_f$

---

In the pseudo-code for Algorithm 3, we begin by initializing a score of zero for each candidate query. Then, for every distinct pair of queries $(c_i, c_j)$, we compare both $(c_i, c_j)$ and $(c_j, c_i)$ to mitigate any order bias, ensuring that both candidates in a pair are fairly evaluated (ignoring the both side comparison will reduce the final performance by roughly 2%). If both queries produce the same execution result on the database, we mark one as the winner and increment its score, as these results suggest consistency. If the execution results differ, we generate a union of the schema used by both queries and use the binary classifier to determine which query is more likely to be correct. The classifier takes into account the question, the two candidate queries, and the combined schema to make its decision. The winner's score is then updated accordingly. After all comparisons, the candidate with the highest score is selected as the final query. In the rare case of a tie in the final scores, we break the tie by selecting one of the candidates arbitrarily.

## 3 EXPERIMENTS

**Datasets and Models** We evaluate the performance of the proposed CHASE-SQL framework on two widely-recognized cross-domain datasets: BIRD (Li et al., 2024c) and Spider (Yu et al., 2018). BIRD contains over 12,751 unique question-SQL pairs from 95 large databases, spanning more than 37 professional domains, with databases designed to resemble real-world scenarios, featuring messy data rows and complex schemas. Spider consists of 10,181 questions and 5,693 unique complex SQL queries across 200 databases, covering 138 domains. The Spider dataset is divided into non-overlapping training, development, and test sets similar to BIRD. For both, we use execution accuracy (EX), the official metric for their respective leaderboard, as the primary evaluation metric to compare methods. Details of the models and their hyperparameters are provided in Appendix section A.3.

**BIRD results** We present the end-to-end Text-to-SQL performance of the proposed CHASE-SQL framework using Claude-3.5-sonnet, Gemini 1.5 pro, and Mistral Large models on the BIRD development set, and Gemini 1.5 pro on the BIRD test set. We compare with both published methods (either with an available codebase and/or paper) and undisclosed methods. For a fair comparison with Gemini 1.5 pro, all LLM calls in the Claude-3.5-sonnet setting, except for the selection model, are made using Claude-3.5-sonnet (previously-trained selection model is reused). To evaluate the

performance of fully open-source models, we used a fine-tuned Qwen2.5-coder model (Team, 2024) as the selection model for the Mistral Large model. As shown in Table 2, CHASE-SQL with Gemini 1.5 pro achieves 73.01% accuracy on the BIRD development set and 73.0% on the BIRD holdout test set, outperforming all previous works and setting a new state-of-the-art performance.

Table 2: Performance Comparison of different Text-to-SQL methods on BIRD benchmark.

| Method | EX (Dev) | EX (Test) |
|---|---|---|
| **Published** | | |
| CHASE-SQL + Gemini 1.5 (Ours) | **73.01** | **73.0** |
| CHASE-SQL + Claude 3.5 Sonnet (Ours) | **69.53** | – |
| CHASE-SQL + Mistral Large (Ours) | **70.33** | – |
| Distillery + GPT-4o (Maamari et al., 2024) | 67.21 | 71.83 |
| CHESS (Talaei et al., 2024) | 65.00 | 66.69 |
| MCS-SQL + GPT-4 (Lee et al., 2024) | 63.36 | 65.45 |
| SuperSQL (Li et al., 2024a) | 58.5 | 62.66 |
| **Undisclosed** | | |
| Insights AI | 72.16 | 70.26 |
| AskData + GPT-4o | 72.03 | 72.39 |
| OpenSearch-v2 + GPT-4o | 69.3 | 72.28 |
| PURPLE-RED + GPT-4o | 68.12 | 70.21 |
| Arcwise + GPT-4o | 67.99 | 66.21 |
| ExSL + granite-34b-code | 67.47 | 67.76 |

Table 3: Performance Comparison of different Text-to-SQL methods on Spider test set.

| Method | EX | Training with Spider |
|---|---|---|
| MCS-SQL + GPT-4 (Lee et al., 2024) | 89.6 | ✓ |
| CHASE-SQL + Gemini 1.5 (Ours) | **87.6** | ✗ |
| CHESS (Talaei et al., 2024) | 87.2 | ✗ |
| DAIL-SQL + GPT-4 (Gao et al., 2023) | 86.6 | ✓ |
| DIN-SQL + GPT-4 (Pourreza & Rafiei, 2024a) | 85.3 | ✓ |
| C3 + ChatGPT (Dong et al., 2023) | 82.3 | ✓ |
| RESDSQL 3B (Li et al., 2023a) | 79.9 | ✓ |
| DIN-SQL + CodeX (Pourreza & Rafiei, 2024a) | 78.2 | ✓ |
| T5-3B+NatSQL (Rai et al., 2023) | 78.0 | ✓ |
| Graphix-3B+PICARD (Li et al., 2023b) | 77.6 | ✓ |

**Spider results** We assess the generalizability of the proposed CHASE-SQL by evaluating it in an end-to-end way on the Spider test set without modifying the few-shot samples in the prompts or training a new selection model, i.e. without using and data from the target distribution. This approach allows us to test the performance of CHASE-SQL on different unseen query and database distributions compared to the data from training distributions. Table 3 demonstrates that CHASE-SQL achieves an execution accuracy of 87.6% on the Spider test set, placing it second among methods that have undergone specific training or prompt optimization for the Spider dataset. This highlights the strong generalizability of CHASE-SQL and its potential for generating high quality Text-to-SQL for unseen samples coming from very different distributions and unique challenges.

## 3.1 GENERATOR AND SELECTION PERFORMANCE

**Generator + Fixer:** To reveal performance of generators, we conducted an ablation study to evaluate the performance of each candidate generation method before and after applying the query fixer using two models of Gemini-1.5-pro and Mistral Large (AI, 2024). We compare the performance of the proposed generators in producing a single candidate query against the original BIRD prompt (Li et al., 2024c), augmented with zero-shot CoT reasoning (Kojima et al., 2022), which serves as the baseline for assessing the quality of prompts. The results, shown in Table 4, indicate that the proposed methods significantly improve SQL generation performance, compared to the naive

Table 4: Ablation studies on single candidate generation performance of the candidate generators compared with original BIRD prompt + zero-shot CoT with Gemini 1.5 pro and Mistral Large on the BIRD dev set.

| | Gemini 1.5 pro | | Mistral Large | |
|---|---|---|---|---|
| Method | EX (%) | Δ(%) | EX (%) | Δ(%) |
| Baseline | 57.75 | - | 54.88 | - |
| QP CoT | 63.62 | +5.87 | 59.64 | 4.76 |
| DC CoT | 63.92 | +6.17 | 58.99 | 4.11 |
| OS ICL | 67.09 | +9.34 | 56.32 | 1.44 |
| Baseline w Query Fixer | 61.58 | +3.83 | 60.03 | 5.15 |
| QP CoT w Query Fixer | 65.51 | +7.76 | 62.64 | 7.76 |
| DC CoT w Query Fixer | 65.77 | +8.02 | 63.75 | 8.87 |
| OS ICL w Query Fixer | 68.02 | +10.27 | 61.47 | 6.59 |

baseline, towards the goal of producing high-quality candidates while maintaining diversity. Among the candidate generators, the online synthetic data generation approach produced an impressive performance of 68.02% with Gemini-1.5-pro model, demonstrating its effectiveness in leveraging test-time compute to improve LLM performance by generating high-quality synthetic examples. Furthermore, the query fixer proved crucial, enhancing the quality of the candidate pool and increasing performance by nearly 2% across all candidate generators.

**Selection:** We conducted an analysis on the binary selection accuracy of the selection agent for cases where, in a pairwise comparison, one candidate is correct and the other is incorrect. We exclude cases where both candidates are either correct or incorrect, as the selection would not affect the outcome since both candidates have the same label. We compare the performance of Claude-3.5-sonnet and Gemini-1.5-pro (both out-of-the-box without fine-tuning) with two fine-tuned models: 1) Gemma 2 9B and 2) Gemini-1.5-flash. As shown in Table 5, both fine-tuned models achieve higher accuracy than the untuned counterparts, demonstrating the importance of fine-tuning to teach the model about the specific preferences.

**Candidate Generation Analysis:** We analyze the performance of each candidate generator method individually. To better understand the performance potential when effectively selecting the correct SQL query from the candidate pool, we generate seven candidate SQL queries from each generator method (21 candidates in total) for all samples in the BIRD development set. We determine this number of candidates based on the observation that increasing the candidate pool beyond 20 did not yield significant improvements, as illustrated in Fig. 2d. By assuming access to an oracle

Table 5: Evaluating the binary selection accuracy of the different selection models.

| Selection Model | Binary Acc. (%) |
|---|---|
| Claude-3.5-sonnet | 60.21 |
| Gemini-1.5-pro | 63.98 |
| Tuned Gemma 2 9B | 64.28 |
| Tuned Gemini-1.5-flash | **71.01** |

selection model that always selects the correct SQL query from the seven candidates, we calculate the upper-bound performance achievable for each generator. Conversely, by assuming an adversarial selection model that always selects the wrong SQL query, we determine the lower-bound performance. Fig. 2 illustrates the upper-bound and lower-bound performance for all three methods together with the performance of our selection agent. As shown, the upper-bound performance of the two different CoT methods is generally higher than that of the synthetic example generation method for different number of candidates. However, their lower-bound performance is also lower than the synthetic method. Lower-bound accuracy reflects cases where all candidates are correct, reducing the noise in the selection process since it doesn't matter which candidate is chosen, so a higher lower-bound is preferred. This is evident in the selection agent's performance, where a drop in the lower bound leads to diminishing returns from increasing the upper bound, causing the selection agent's performance to plateau. Additionally, the upper-bound performance of combining all three methods reaches **82.79%**, highlighting the significant room for improvement through better candidate picking methods. This demonstrates that the LLM's parametric knowledge already contains the information needed to solve most questions, highlighting the need for ensemble approaches to effectively extract and utilize this knowledge.

Additionally, we evaluate the upper-bound performance by combining all candidates from three candidate generation methods across the simple, moderate, and challenging difficulty levels for the BIRD development set. These difficulty categories are assigned by human experts during the creation of the BIRD development set. Fig. 2d shows that, as expected, the upper-bound performance increases with the number of candidates across all difficulty levels. However, for the challenging and moderate classes, the improvement plateaus earlier than in the simple class, suggesting that generating more samples does not further improve the upper-bound performance for these two difficulty levels.

Fig. 2 presents a Venn diagram showcasing the performance of three generation methods: Query Plan, Divide and Conquer, and with Synthetic Examples. The numbers within the intersecting regions represent the instances where multiple methods generated at least one correct candidate. This diagram visually highlights the unique contributions of each method, which indicates the necessity of using all three generators. Additionally, in Fig. 3b, we compare the number of correct queries generated by each SQL generation method that are not correct by the other generators. The divide-and-conquer approach outperforms the others on challenging questions, while the query plan method excels on moderately difficult queries. To further analyze the performance of the generators across different domains and varying numbers of columns and tables, we compare the number of correct queries generated for each database, as shown in Appendix Fig. 5. As illustrated, both CoT methods generally perform similarly across databases, while the online synthetic example generation method significantly increases diversity, resulting in more correct answers overall across different databases.

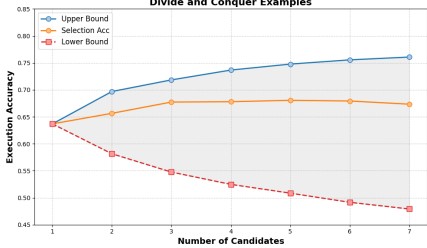

(a) Upper-bound and lower-bound Accuracy for Divide and Conquer CoT

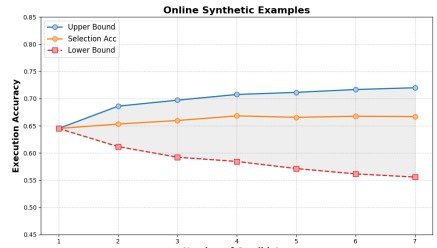

(b) Upper-bound and lower-bound Accuracy for Online Synthetic Example

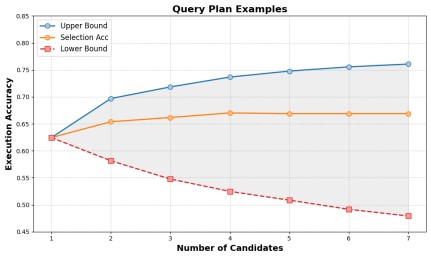

(c) Upper-bound and lower-bound performance for Query Plan CoT.

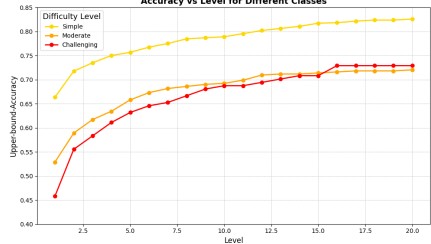

(d) Upper-bound performance of all three candidate generators across different difficulty categories.

Figure 2: Comparison of the upper- and lower-bound performance of different candidate generators.

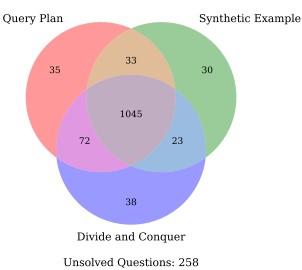

(a) Venn diagram illustrating the number of instances for which each method: Query Plan, Synthetic Example, Divide and Conquer, produces at least one correct candidate. The overlap regions represent multiple methods generating correct candidates.

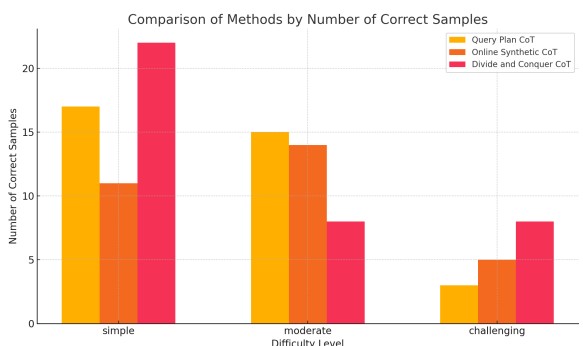

(b) Number of correct queries across different complexity levels that were answered by each method.

Figure 3: Comparison of SQL generation methods: Venn diagram showing unique and overlapping correct answers (left) and the performance across different complexity levels (right).

**Selection Agent Analysis:** We evaluate the query-picking performance by comparing the Text-to-SQL execution accuracy of the selection agent with the self-consistency method (using majority voting) Wang et al. (2022), an oracle model (upper bound), and an adversarial model (lower bound). To conduct the evaluation, we generate 10 samples from each candidate generation method using two different sampling temperatures: 0.5 and 1.8. The results, shown in Table 6, demonstrate that the selection agent significantly outperforms the self-consistency method with a large margin, roughly **6%**. As expected, increasing the sampling temperature raises the upper bound but also lowers the lower bound. This effect is more pronounced for the synthetic data generation method compared to the two CoT methods, mainly because LLMs generate reasoning steps before producing the final SQL query, which helps mitigate the randomness introduced by high-temperature sampling. The performance with self-consistency method generally decreases as temperature increases, since the

majority cluster becomes smaller with more random queries. However, the proposed trained selection agent is less affected by temperature scaling and, in two cases, even improved its performance with a more diverse pool of samples.

Table 6: Performance comparison of different picking methods on the candidates generated by the candidate generators on BIRD development set with two different temperatures. QP refers to query plan COT, DC refers to divide and conquer COT, and OS is the online synthetic example generation method.

| Picking Method | QP (T=0.5) | QP (T=1.8) | DC (T=0.5) | DC (T=1.8) | OS (T=0.5) | OS (T=1.8) |
|---|---|---|---|---|---|---|
| Lower Bound | 50.46 | 48.63 | 51.37 | 47.39 | 60.43 | 50.98 |
| Upper Bound | 78.55 | 80.44 | 78.42 | 79.34 | 74.77 | 79.66 |
| Self-consistency | 65.78 | 65.51 | 66.43 | 64.41 | 67.34 | 66.88 |
| Our Selection Agent | 71.7 | 71.73 | 71.31 | 70.53 | 70.4 | 71.38 |

## 3.2 ABLATION STUDIES

In the previous sections, we evaluate the importance of the selection agent and each candidate generation method. Next, we focus on the analysis of the remaining components of CHASE-SQL: LSH for value retrieval, the query fixer, and three reasoning strategies (QP, OS, and DC). Table 7 shows the performance of CHASE-SQL without each of these steps, highlighting their significance in achieving higher-quality performance. The results demonstrate the contribution of each component, where

Table 7: Ablation studies on the performance of CHASE-SQL after removing the query fixer, LSH for value retrieval, and reasoning strategies, i.e., QP, OS, and DC.

| Method | Execution Accuracy (%) | $\Delta(\%)$ |
|---|---|---|
| CHASE-SQL All | 73.01 | - |
| CHASE-SQL w self-consistency | 68.84 | -4.17 |
| CHASE-SQL w ranker agent | 65.51 | -7.5 |
| CHASE-SQL w/o LSH | 70.09 | -2.92 |
| CHASE-SQL w/o Query Fixer | 69.23 | -3.78 |
| CHASE-SQL w/o QP | 72.36 | -0.65 |
| CHASE-SQL w/o OS | 72.16 | -0.85 |
| CHASE-SQL w/o DC | 71.77 | -1.24 |

removing LSH, the query fixer, or any of the candidate generators leads to a reduction in execution accuracy, further validating the importance of these components of CHASE-SQL. Moreover, the table compares the performance of our binary selection agent with two other selection methods: self-consistency (Wang et al., 2022) and a ranker agent. The ranker agent receives all candidates generated by our three candidate generators in a single prompt, compares them, and produce a ranking for each. For the ranker agent, we select the query with the lowest rank as the best answer. The binary selection agent significantly outperforms both the self-consistency and ranker agents, demonstrating the effectiveness of the proposed method.

## 4 CONCLUSION

We introduce a novel agentic framework, CHASE-SQL, to leverage test-time compute for generating diverse, high-quality SQL queries and accurately selecting the correct one. We propose multiple chain-of-thought prompting methods and an online synthetic example generation technique, along with a query selection mechanism that scores candidates based on pairwise comparisons. Our framework, CHASE-SQL, sets a new state-of-the-art in the notable public Text-to-SQL leaderboard (at the time of the submission), demonstrating the effectiveness of test-time computation for both generating diverse queries and selecting the most accurate response. CHASE-SQL addresses key issues like query diversity and selection optimization, paving the way for further improvements in complex reasoning tasks encountered at real-world Text-to-SQL challenges.

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

# A APPENDIX

## A.1 LIMITATIONS AND FUTURE WORKS

Based on the analysis presented in this paper, we demonstrate that the parametric knowledge of recent large language models, such as Gemini-1.5-pro and Mistral Large, contains the necessary information to answer most challenging questions in notable text-to-SQL benchmarks, as evidenced by their high pass@K performance. This highlights the challenge of effectively utilizing the reasoning ability of these models to select the best answer among the candidates. In our work, we concluded that pairwise comparison is an effective approach to identify the best candidate. However, we believe this performance could be further improved by leveraging the reasoning capabilities of the models, either through chain-of-thought prompting or employing more sophisticated search methods, which we leave as directions for future work. Additionally, further research is needed to address the detection of ambiguous questions and to improve the reliability of text-to-SQL systems. Most methodologies in this domain assume that all questions are answerable, which remains a significant limitation in current text-to-SQL approaches.

## A.2 RELATED WORKS

Early Text-to-SQL methods predominantly utilized sequence-to-sequence architectures, encoding user queries and database schemas using models such as Graph Neural Networks (GNNs), Recurrent Neural Networks (RNNs), Long Short-Term Memory (LSTM) networks, and pre-trained transformer encoders (Hwang et al., 2019; Cai et al., 2021; Cao et al., 2021). On the decoding side, these systems employed either slot-filling or auto-regressive modelling approaches to construct the final SQL queries from the encoded inputs (Choi et al., 2021; Wang et al., 2019). Additionally, tabular language models like TaBERT (Yin et al., 2020), TaPas (Herzig et al., 2020), and Grappa (Yu et al., 2020) have been developed to encode both tables and textual data effectively. However, the landscape has evolved with the widespread use of LLMs, which have largely replaced earlier methods with their superior performance (Katsogiannis-Meimarakis & Koutrika, 2023; Quamar et al., 2022). Initially, efforts concentrated on optimizing prompt designs for these LLMs (Pourreza & Rafiei, 2024a; Gao et al., 2023; Dong et al., 2023). Subsequent advancements have introduced more complex methodologies, including schema linking (Li et al., 2024b; Talaei et al., 2024; Pourreza & Rafiei, 2024a;b), self-correction or self-debugging (Chen et al., 2023; Wang et al., 2023; Talaei et al., 2024), and self-consistency techniques (Lee et al., 2024; Sun et al., 2023; Talaei et al., 2024; Maamari et al., 2024), further enhancing the performance by proposing complex LLM-based pipelines.

As previously discussed, one approach to enhance Text-to-SQL performance is based on the consistency of LLM responses. The self-consistency approach, as proposed by Wang et al. (2022), involves sampling multiple responses from an LLM and selecting the most consistent answer based on the majority vote. In the Text-to-SQL context, this technique extends to generating multiple SQL queries for a given question, grouping these queries by their execution results, and selecting a query from the largest cluster as the most consistent answer (Gao et al., 2023; Sun et al., 2023; Talaei et al., 2024). However, recent studies have pointed out the limitations of this method in reliably identifying the correct answer. In response, MCS-SQL (Lee et al., 2024) introduced an approach that utilizes an LLM to rerank the most consistent answers, moving beyond simple majority voting. Despite these advancements, reliance on consistency as a filtering mechanism can inadvertently exclude correct queries that are less frequent among generated candidates, as a critical bottleneck.

## A.3 MODELS

All experiments are conducted using models from the Gemini and Claude, known for their ability to handle long contextual information (Maamari et al., 2024), which is crucial for the Text-to-SQL task involving queries from large databases. For candidate generation, online synthetic example generation, query fixing, column filtering, and keyword extraction, we reported the performance with two models of Gemini 1.5 Pro and Claude-3.5-Sonnet. For the query-picking model, we train a Gemini 1.5 Flash model (which has much less latency than the Gemini 1.5 Pro model) on a dataset of 3.8K samples generated by running the candidate generators on the BIRD training dataset. The

Gemini 1.5 Flash model is trained for 10 epochs using a LoRA adapter with a rank of 16 using Vertex AI tuning API.

## A.4    QUERY FIXING ALGORITHM

In this section, we present the algorithm of the query fixing step of our propose approach.

---

**Algorithm 4** Query fixing method.

---

**Require:**   Set of candidate SQL queries $C = \{c_1, c_2, ..., c_n\}$, user question $Q_u$, hint $H_u$, target database $D$,
   max query fixing threshold $\beta$, and a fixer model $\theta_f$
1:  $C_{fixed} \leftarrow \emptyset$ // *Initialize an empty set $C_{fixed}$ to store the fixed queries*
2:  **for** each candidate query $c_i \in C$ **do**
3:     $Execution_i \leftarrow Execute(c_i, D)$ // *Execute the SQL query over the database*
4:     **if** $error$ in $Execution_i$ or $Execution_i = []$ **then**
5:        **for** $j \in \{1, 2, ..., \beta\}$ **do** // *Try fixing up to $\beta$ times*
6:           $c_i \leftarrow \theta_f(D, c_i, Execution_i)$ // *Fix the query using the fixer model*
7:           $Execution_i \leftarrow Execute(c_i, D)$ // *Re-execute the fixed query*
8:           **if** not ($error$ in $Execution_i$ or $Execution_i = []$) **then**
9:              $C_{fixed} \leftarrow C_{fixed} \cup \{c_i\}$ // *Add the fixed query to the set*
10:             **break** // *Exit the fixing loop if successful*
11:          **end if**
12:       **end for**
13:    **else**
14:       $C_{fixed} \leftarrow C_{fixed} \cup \{c_i\}$ // *Add the query as is if no fixing is needed*
15:    **end if**
16: **end for**
17: **return** $C_{fixed}$ // *Return the set of fixed queries*

---

A.5    VALUE RETRIEVAL EXAMPLE

In this section, we provide an example of the value retrieval step. For the Given Question: "What is the highest eligible free rate for K-12 students in the schools in Alameda County?", from the "california_schools" Database, the closest database values that are retrieved from the LSH and after reranking are as follows:

Table "schools":
"SOCType": ["Preschool"],
"EILName": ["Preschool"],
"School": [
"Preschool",
"MethodSchools",
"Alameda County Community",
"Alameda County Opportunity",
"Alameda High"],
"MailStreet": ["4600 Student Lane"],
"Street": ["4600 Student Lane"],
"AdmLName1": ["Free"],
"AdmLName2": ["Freeman"],
"MailCity": ["Alameda"],
"City": ["Alameda"],
"AdmFName1": ["Kate", "Nate", "Bree"],
"GSserved": ["K-12"],
"GSoffered": ["K-12"],
"StreetAbr": ["4600 Student Ln."],
"MailStrAbr": ["4600 Student Ln."],
"AdmLName3": ["Yount"],
"AdmFName3": ["Bree"],
"County": ["Alameda"],
"District": ["Alameda Unified", "Tri-County ROP"],
Table "frpm":
"School Type": ["Preschool"],
"School Name": [
"MethodSchools",
"Alameda County Community",
"Alameda High"
], "County Name": ["Alameda"],
Table "satscores":
"sname": ["Alameda High"],
"cname": ["Alameda"],
"dname": ["Alameda County Office of Education"]

Figure 4: An example of the divide and conquer CoT method

A.6    PERFORMANCE BASED ON DATABASE

In this section, we present the number of samples across different databases where only one of the candidate generators produces a correct result, meaning the other two generators fail to provide a correct answer. A value of zero for any generator in this figure indicates that whenever that generator produces a correct result, the other two generators also manage to generate at least one correct answer.

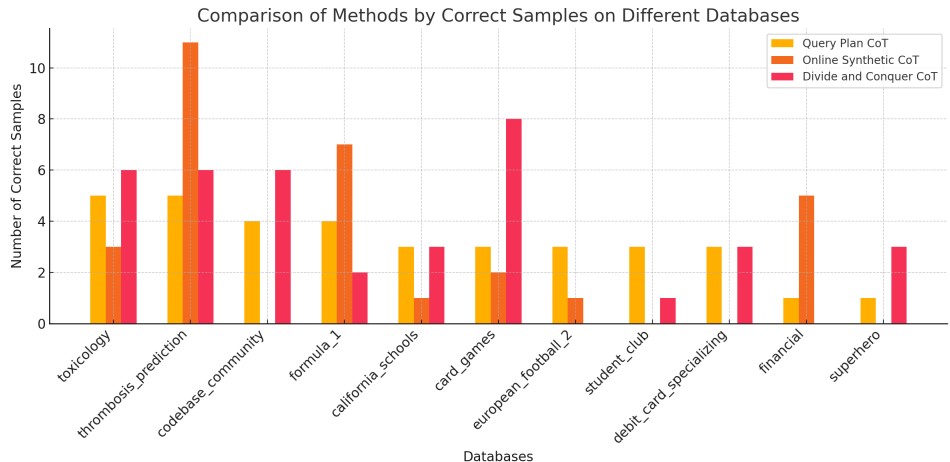

Figure 5: Number of correct queries by each method across different databases of BIRD development set.

## A.7  ERROR ANALYSIS

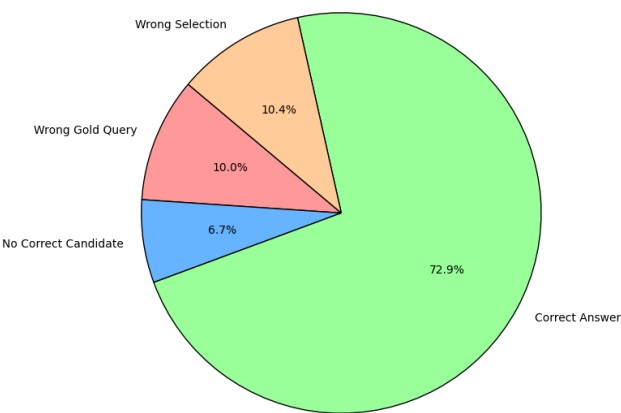

Figure 6: Distribution of system performance based on the final answer correctness. The chart shows the proportion of correct final answers, correct queries existing among candidates but not chosen (wrong selection), no correct candidate cases, and cases were the golden SQL query is wrong.

Fig. 6 provides a pie chart that breaks down the system's performance into four categories: correct final answer (72.9%), correct exists among candidates but not chosen (10.4%), wrong generations or no correct candidate (6.7%), and wrong golden query (10.0%). The majority of responses are correct final answers, but a notable portion falls under correct answers not being chosen by the system. This breakdown helps in understanding areas where the system excels and where improvements can be targeted.

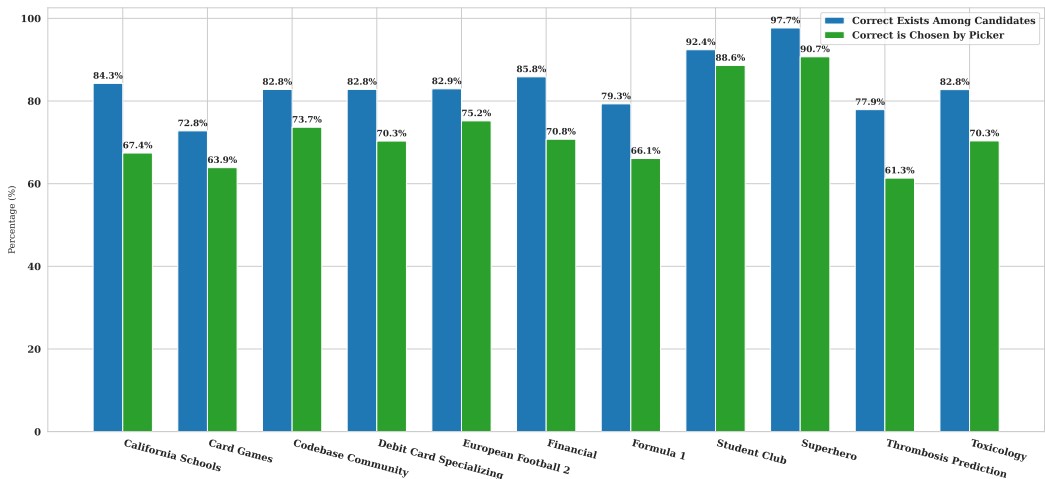

Figure 7: Correctness comparison of the system across different databases in two metrics: (1) percentage where the correct query exists among the candidates, and (2) percentage where the correct query is chosen by the selection agent.

Fig. 7 presents a comparative analysis of system correctness across multiple databases. The x-axis lists various databases or categories such as California Schools, Formula 1, and Superhero, while the y-axis represents the percentage performance. Two key metrics are visualized: the first is the percentage where the correct answer exists among the candidates (shown by one bar per category), and the second is the percentage where the correct answer is chosen by the selection system (depicted by a second bar for each category).

### A.7.1 SELECTION AGENT ERROR ANALYSIS

In this section, we examine cases where at least one of the candidate SQL queries generated by the three generators matched the ground truth answer, but the selection agent assigned the highest score to another, incorrect candidate. We categorized these errors into four groups: (1) Vague questions, (2) Wrong picking, (3) Data Integrity Error, and (4) Incorrect gold query. Fig. 8 illustrates the distribution of each category among the sample queries. In the following sections, we will discuss each of these categories in more detail.

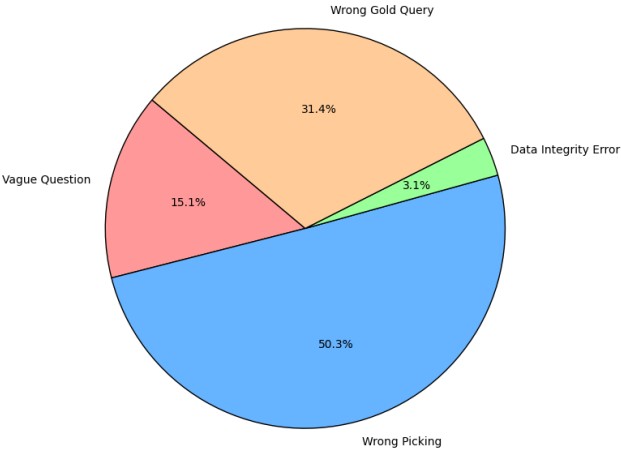

Figure 8: Error analysis on the cases where selection agent failed to pick the correct SQL query which was among the candidates.

**Wrong Picking Errors:** The largest portion of errors occurs when the candidate with the highest score from the selection agent is missing a required column, table, or SQL clause. In our analysis, we could not identify a specific types of patterns in the model's mistakes as these mistakes includes different types of the errors almost for each instance. Fig. 9 provides an example where the selected SQL query incorrectly uses * to return all columns, instead of just returning the id as specified in the ground truth answer.

Figure 9: An example of selection agent preferred SQL query which is incorrect.

**Wrong Golden Query Error:** The second largest portion of errors occurs when the ground truth SQL query is incorrect, and one of the candidate queries generated by our model replicates the same mistake. However, the selection agent ultimately picks another candidate that correctly answers the question. Fig. 10 provides an example of such a case, where the ground truth query includes an extra molecule_ID column in the SELECT clause, which was not specified in the question.

Figure 10: An example of an error case where the selection agent picked a correct SQL query and the gold query was wrong.

**Vague Question:** Another significant portion of errors occurs when the question does not specify which column to return or use for filtering, and multiple columns could satisfy the query. In these cases, although one of the candidates was the correct SQL query, the selection model favored another response that could also be considered correct. Fig. 11 illustrates such a case where "Fresno" could refer to either a city or a county, but the question doesn't specify which one to return. The selection model chose the query that used "city" and did not select the candidate that used "county.

Figure 11: An example of an error case where the selection model picked a query which could be considered as correct as the question is vague.

**Data Integrity Error:** Finally, the smallest category of errors involves cases where two or more columns are supposed to have consistent values, but one or more columns contain missing values.

For example, Fig. 12 shows a case where the "School" and "School Name" columns were both expected to contain the names of schools, but one of the columns has missing values.

> **Question:** List the names of schools with more than 30 difference in enrollments between K-12 and ages 5-17? Please also give the full street adress of the schools
>
> **Evidence:** Diffrence in enrollement = `Enrollment (K-12)` - `Enrollment (Ages 5-17)`
>
> **Gold SQL:** SELECT T1.School, T1.Street FROM schools AS T1 INNER JOIN frpm AS T2 ON T1.CDSCode = T2.CDSCode WHERE T2.`Enrollment (K-12)` - T2.`Enrollment (Ages 5-17)` > 30
>
> **Picked SQL:** SELECT T2.`School Name`, T1.Street FROM schools AS T1 INNER JOIN frpm AS T2 ON T1.CDSCode = T2.CDSCode WHERE T2.`Enrollment (K-12)` - T2.`Enrollment (Ages 5-17)` > 30

Figure 12: An example of an error case where the selection agent picked a correct candidate but because of the data inconsistency the execution accuracy was zero for this candidate.

## A.7.2 ERROR ANALYSES

We present the manual error analysis we conducted on one-third of the cases where none of the generated candidate queries were correct. We categorized these errors into five main types: (1) Schema linking errors, (2) Incorrect logic, (3) SQL function errors, (4) JOIN issues, and (5) Ignoring evidence. Fig. 13 illustrates the distribution of these error categories. As shown, the most common errors occur when none of the candidate queries correctly utilized the columns or tables required to answer the question. In the following section, we describe the specific types of errors that fall under each category.

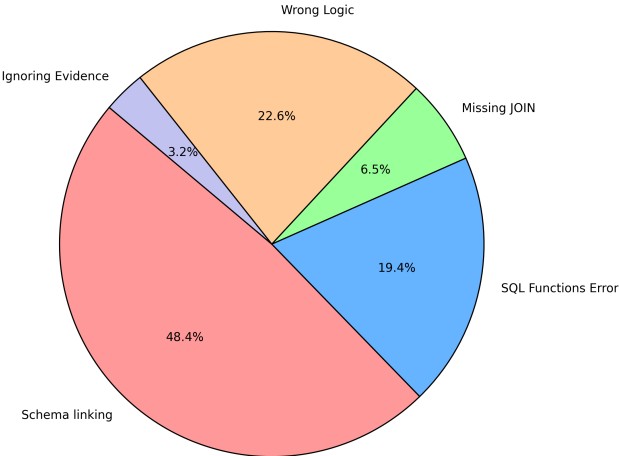

Figure 13: Error analysis on the cases where all candidate generators failed to produce a single correct answer.

**Schema Linking Errors:** The schema linking errors category includes cases where none of the generated candidate SQL queries correctly use the columns required to answer the question. These errors often occur in databases where column names are ambiguous or confusing for the model. Fig. 14 provides an example where the LLM failed to correctly calculate the average to return the correct column that was expected.

**Wrong Logic Error:** This category, which represents the second largest portion of errors, includes cases where the logic of the generated candidate queries is incorrect. These errors involve missing elements such as the DISTINCT keyword, NOT NULL conditions, missing columns in the SELECT clause, or incorrect or missing conditions in the WHERE or HAVING clauses. An example is shown in Fig. 15, provides an example where the LLM failed to correctly calculate the average total price due to incorrect logic computing the average total price.

Figure 14: An example of schema linking error category.

Figure 15: An example of wrong logic error category.

**SQL Functions Error:** This category, the third-largest source of errors, includes queries where the error results from the incorrect use of, or failure to include, SQL functions such as COUNT(), CAST(), AVG(), ROUND(), and others. Fig. 16 illustrates a case where none of the candidate queries used the ROUND() function as required by the question.

Figure 16: An example of wrong usage of SQL functions error category.

**Joins Error:** This category represents a smaller portion of the error cases, where the source of the error is missing one or more tables in the JOIN clauses of the predicted SQL queries.

**Ignoring Evidence Error:** This category represents the smallest portion of error cases, where the correct column or table is explicitly mentioned in the provided evidence, but the LLM disregards it and uses its own selection instead.

## A.8 DIVIDE AND CONQUER PROMPT

In this section, we provide an example of a divide-and-conquer prompt used in one of the few-shot in-context learning demonstrations to illustrate the decomposition and aggregation steps.

Database Info
{DATABASE_SCHEMA}
**************************
Question
Question: What is the gender of the youngest client who opened account in the lowest average salary branch?
Hint: Given that Later birthdate refers to younger age; A11 refers to average salary

**************************
**1. Divide and Conquer:**

* **Main Question:** What is the gender of the youngest client who opened account in the lowest average salary branch?
* **Analysis:** Question asking about 'gender', and it appears in table 'client'. We will use this as the output column, selecting it from the youngest client in the lowest average salary branch.
* **Pseudo SQL:** SELECT 'T1'.'gender' FROM 'client' AS 'T1' WHERE youngest client in the lowest average salary branch

* **Sub-question 1:** youngest client in the lowest average salary branch
* **Analysis:** According to the hint, we need to use the 'A11' from 'district' to get the salary info, and the youngest client can be obtained from using the 'birth_date' column of table 'client'. The items between these two tables can be INNER JOIN using district_id.
* **Pseudo SQL:** SELECT 'T1'.'client_id' FROM 'client' AS 'T1' INNER JOIN 'district' AS 'T2' ON 'T1'.'district_id' = 'T2'.'district_id' WHERE lowest average salary branch ORDER BY 'T1'.'birth_date' DESC NULLS LAST LIMIT 1

* **Sub-question 1.1:** lowest average salary branch
* **Analysis:** We can get the lowest average salary branch using order by 'A11' ASC and pick top 1. The column 'A11' is not NULLABLE, so we do not need to add "IS NOT NULL" filter * **Pseudo SQL:** SELECT 'district_id' FROM 'district' ORDER BY 'A11' ASC LIMIT 1

**2. Assembling SQL:**

* **Sub-question 1.1 (lowest average salary branch):** * **SQL:** SELECT 'district_id' FROM 'district' ORDER BY 'A11' ASC LIMIT 1

* **Sub-question 1 (youngest client in the lowest average salary branch):**
* **SQL:** SELECT 'T1'.'client_id' FROM 'client' AS 'T1' INNER JOIN 'district' AS 'T2' ON 'T1'.'district_id' = 'T2'.'district_id' WHERE 'T2'.'district_id' IN (SELECT 'district_id' FROM 'district' ORDER BY 'A11' ASC LIMIT 1) ORDER BY 'T1'.'birth_date' DESC NULLS LAST LIMIT 1

* **Main Question (gender of the client):**
* **SQL:** SELECT 'T1'.'gender' FROM 'client' AS 'T1' WHERE 'T1'.'client_id' = (SELECT 'T1'.'client_id' FROM 'client' AS 'T1' INNER JOIN 'district' AS 'T2' ON 'T1'.'district_id' = 'T2'.'district_id' WHERE 'T2'.'district_id' IN (SELECT 'district_id' FROM 'district' ORDER BY 'A11' ASC LIMIT 1) ORDER BY 'T1'.'birth_date' DESC NULLS LAST LIMIT 1)

**3. Simplification and Optimization:**
* The nested queries can be combined using a single 'INNER JOIN' and the filtering can be done within a single 'ORDER BY' clause.

**Final Optimized SQL Query:**
SELECT 'T1'.'gender' FROM 'client' AS 'T1' INNER JOIN 'district' AS 'T2' ON 'T1'.'district_id' = 'T2'.'district_id' ORDER BY 'T2'.'A11' ASC, 'T1'.'birth_date' DESC NULLS LAST LIMIT 1

Figure 17: An example of the divide and conquer CoT method

An example of question/SQL pair where the divide-and-conquer CoT approach was the only generator that solved the problem.

```sql
Question: How much faster in percentage is the champion than the driver
who finished the race last in the 2008 Australian Grand Prix?

Query:
WITH time_in_seconds AS (
  SELECT
    T1.positionOrder,
    CASE
      WHEN T1.positionOrder = 1 THEN
        (CAST(SUBSTR(T1.time, 1, 1) AS REAL) * 3600) +
        (CAST(SUBSTR(T1.time, 3, 2) AS REAL) * 60) +
        CAST(SUBSTR(T1.time, 6) AS REAL)
      ELSE
        CAST(SUBSTR(T1.time, 2) AS REAL)
    END AS time_seconds
  FROM
    results AS T1
  INNER JOIN
    races AS T2
    ON T1.raceId = T2.raceId
  WHERE
    T2.name = 'Australian Grand Prix'
    AND T1.time IS NOT NULL
    AND T2.year = 2008
),
champion_time AS (
  SELECT time_seconds
  FROM time_in_seconds
  WHERE positionOrder = 1
),
last_driver_incremental AS (
  SELECT time_seconds
  FROM time_in_seconds
  WHERE positionOrder = (
    SELECT MAX(positionOrder)
    FROM time_in_seconds
  )
)
SELECT
  (CAST((SELECT time_seconds FROM last_driver_incremental) AS REAL) *
100) /
  (SELECT time_seconds +
    (SELECT time_seconds FROM last_driver_incremental)
  FROM champion_time);
```

Figure 18: An example of query that has been solved only with the divide and conquer CoT method.

## A.9   QUERY PLAN PROMPT

In this section, we provide an example of a query (execution) plan prompt used in one of the few-shot in-context learning demonstrations to illustrate the steps.

Database Info
{DATABASE_SCHEMA}
**************************
Answer Repeating the question and evidence, and generating the SQL with a query plan.
**Question**: How many Thai restaurants can be found in San Pablo Ave, Albany?
**Evidence**: Thai restaurant refers to food_type = 'thai'; San Pablo Ave Albany refers to street_name = 'san pablo ave' AND T1.city = 'albany'

**Query Plan**:
** Preparation Steps:**
1. Initialize the process: Start preparing to execute the query.
2. Prepare storage: Set up storage space (registers) to hold temporary results, initializing them to NULL.
3. Open the location table: Open the location table so we can read from it.
4. Open the generalinfo table: Open the generalinfo table so we can read from it.

** Matching Restaurants:**
1. Start reading the location table: Move to the first row in the location table.
2. Check if the street matches: Look at the street_name column of the current row in location. If it's not "san pablo ave," skip this row.
3. Identify the matching row: Store the identifier (row ID) of this location entry.
4. Find the corresponding row in generalinfo: Use the row ID from location to directly find the matching row in generalinfo.
5. Check if the food type matches: Look at the food_type column in generalinfo. If it's not "thai," skip this row.
6. Check if the city matches: Look at the city column in generalinfo. If it's not "albany," skip this row.

** Counting Restaurants:**
1. Prepare to count this match: If all checks pass, prepare to include this row in the final count.
2. Count this match: Increment the count for each row that meets all the criteria.
3. Move to the next row in location: Go back to the location table and move to the next row, repeating the process until all rows are checked.
4. Finalize the count: Once all rows have been checked, finalize the count of matching rows.
5. Prepare the result: Copy the final count to prepare it for output.

** Delivering the Result:**
1. Output the result: Output the final count, which is the number of restaurants that match all the specified criteria.
2. End the process: Stop the query execution process.
3. Setup phase: Before starting the actual query execution, the system prepares the specific values it will be looking for, like "san pablo ave," "thai," and "albany."

**Final Optimized SQL Query:**

SELECT COUNT(T1.id_restaurant) FROM generalinfo AS T1 INNER JOIN location AS T2 ON T1.id_restaurant = T2.id_restaurant WHERE T1.food_type = 'thai' AND T1.city = 'albany' AND T2.street_name = 'san pablo ave'

Figure 19: An example of the query plan CoT method

Fig. 20 provides an example of a query plan generated by using the EXPLAIN command.

Figure 20: An example of SQLite query plan generated by using the EXPLAIN command.

Additionally Fig. 21 provides an example question that was solved by using the query plan-based CoT strategy.

Figure 21: An example of query that has been solved only with the query plan CoT method.

## A.10 QUERY FIXING PROMPT

In this section, we provide the prompt template for the SQL query fixing step.

**Task Description:**
You are an SQL database expert tasked with correcting a SQL query. A previous attempt to run a query did not yield the correct results, either due to errors in execution or because the result returned was empty or unexpected. Your role is to analyze the error based on the provided database schema and the details of the failed execution, and then provide a corrected version of the SQL query.

**Procedure:**
1. Review Database Schema:
- Examine the table creation statements to understand the database structure.
2. Analyze Query Requirements:
- Original Question: Consider what information the query is supposed to retrieve.
- Hint: Use the provided hints to understand the relationships and conditions relevant to the query.
- Executed SQL Query: Review the SQL query that was previously executed and led to an error or incorrect result.
- Execution Result: Analyze the outcome of the executed query to identify why it failed (e.g., syntax errors, incorrect column references, logical mistakes).
3. Correct the Query:
- Modify the SQL query to address the identified issues, ensuring it correctly fetches the requested data according to the database schema and query requirements.

**Output Format:**

Present your corrected query as a single line of SQL code, after Final Answer. Ensure there are no line breaks within the query.

Here are some examples:
{EXAMPLES}
======= Your task =======
**************************
Table creation statements
{DATABASE_SCHEMA}
**************************
The original question is:
Question:
{QUESTION}
Evidence:
{HINT}
The SQL query executed was:
{QUERY}
The execution result:
{RESULT}
**************************
Based on the question, table schema and the previous query, analyze the result try to fix the query.

Figure 22: The prompt template used for query fixing

## A.11   SELECTION AGENT PROMPT

In this section, we provide the prompt template used for training and query picking at test time by the trained selection agent. Note that the database schema used in this step is the union of the columns and tables by the two candidates instead of using the full-schema of all tables in the database.

Instruction:
Given the DB info and question, there are two candidate queries. There is correct one and incorrect one, compare the two candidate answers, analyze the differences of the query and the result. Based on the original question and the provided database info, choose the correct one.
*************************
Database Schema
{DATABASE_SCHEMA}
*************************
Question:
{QUESTION}
Evidence:
{HINT}
*************************
Candidate A
{CANDIDATE_A_QUERY}
Execution result
{CANDIDATE_A_RESULT}
*************************
Candidate B
{CANDIDATE_B_QUERY}
Execution result
{CANDIDATE_B_RESULT}

Just output the correct answer "A" or "B".

Figure 23: The prompt template used for query fixing

## A.12 GENERATED SYNTHETIC EXAMPLES ANALYSIS

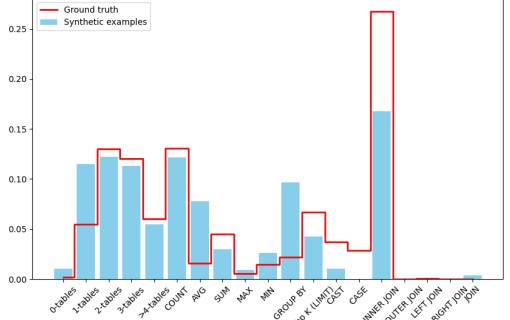

**Question:** What is the highest eligible free rate for K-12 students in the schools in Alameda County?
**SQL:** SELECT `Free Meal Count (K-12)` / `Enrollment (K-12)`
**FROM** frpm
**WHERE** `County Name` = 'Alameda'
**ORDER BY** (CAST(`Free Meal Count (K-12)` AS REAL) / `Enrollment (K-12)`)
**DESC LIMIT** 1

**Common SQL feature example:**
**input:** What is the total enrollment of schools in districts with an average SAT math score above 600?
**output:** SELECT SUM(T2.`Enrollment (K-12)`) FROM satscores AS T1 INNER JOIN frpm AS T2 ON T1.cds = T2.CDSCode WHERE T1.AvgScrMath > 600

**Simple example with filtered columns:**
**input:** What is the average `Enrollment (K-12)` for schools in Alameda County?
**output:** SELECT AVG(`Enrollment (K-12)`) FROM frpm WHERE `County Name` = 'Alameda'

**input:** Which school in Alameda County has the highest `Free Meal Count (K-12)`?
**output:** SELECT School FROM schools AS T1 INNER JOIN frpm AS T2 ON T1.CDSCode = T2.CDSCode WHERE T1.County = 'Alameda' ORDER BY T2.`Free Meal Count (K-12)` DESC LIMIT 1

Figure 24: Synthetic examples generated for the 'california_schools' database question with different guidelines for common SQL features and filtered columns.

Table 8: Ablation studies on synthetic example generation guidelines, $R_f$ with common SQL features and $R_t$ with filtered schema. The baseline is the original BIRD prompt Zero-shot CoT with Gemini 1.5 pro on the BIRD dev set. Total of 75 examples are generated for each example set ($R_f$, $R_t$) and for the mixed ($R_f + R_t$)

| Method | Execution Accuracy (%) | $\Delta(\%)$ |
|---|---|---|
| Baseline (Zero-shot) | 57.75 | - |
| OS w/ $R_f$ | 65.45 | +7.7 |
| OS w/ $R_t$ | 66.75 | +9.0 |
| OS w/ $R_f + R_t$ | 67.09 | +9.34 |

Table 8 illustrates the ablation studies done with different guidelines and their generated example sets. Compared to the baseline (no example), the user question and its associated data schema targetted synthetic examples can help; we try to promote the diversity of the examples to avoid overfitting the output to certain patterns (e.g., the model always writes a SQL with JOIN if shown mostly JOIN examples).

Figure 25: Distribution (normalized) of synthetic vs. ground truth examples in different SQL features/clauses. All examples are generated using *gemini-1.5-pro* for the questions and schemas from the BIRD-Bench dev dataset.

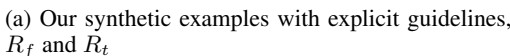

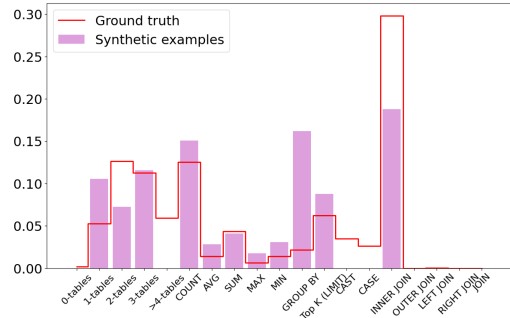

(a) Our synthetic examples with explicit guidelines, $R_f$ and $R_t$

(b) Synthetic examples following cross-domain data augmentation strategy (Li et al., 2024b)

Fig. 25a shows the SQL feature distribution of the generated synthetic examples for the BIRD dev dataset, which closely follows the actual SQL features distribution, except CASE statement. We

omit CASE statement examples, since showing examples with CASE statement did not help with the generation, unless the ground-truth SQL query actually used it.

Table 9: Example quality comparison between our synthetic examples (OS) and other example sets, prepared by two other approaches: relevant training examples by question similarity ($\sigma$(train)) and a different synthetic example generation strategy (CodeS). To isolate the impact of example quality, we only generated a single output candidate without any self-correction with varying numbers of examples $n$.

| $n$ | 5 | 25 | 75 | 125 |
|---|---|---|---|---|
| $\sigma$(train) | 58.80 | 58.54 | 57.69 | 56.91 |
| CodeS | 60.76 | 60.63 | 59.97 | 59.06 |
| OS | 62.13 | 64.02 | 64.41 | 63.69 |

Table 9 demonstrates how the online synthetic example generation (OS) yields more useful example set for in-context learning, compared to two other approaches. Training dataset is a commonly used source of few-shot examples, where examples are selected by question similarity ($\sigma$(train)). Data augmentation is another technique used for cross-domain adaptation, where either extra fine-tuning data or few-shot examples are synthesized.The proposed technique in (Li et al., 2024b) uses two-step example generation, where the model is 1) asked to come up with questions to ask given the schema; 2) asked to fill in the blanks of example templates with the schema elements. There is no specific guidelines to the example and SQl structures for the first step, and the second step uses a set of universal question/SQL templates with limited complexity (e.g., only single table, non-nested queries with up to 3 columns). The resulting example SQL feature distribution is shown in Fig. 25b.

### A.13 SYNTHETIC EXAMPLE GENERATION PROMPTS

In this section we provided the prompt template for the online synthetic example generation step.

You are a SQLite SQL expert. Your job is to create {k} examples, where each example consists of a question and a SQL query to fetch the data for it. I want each example to look like this, question input and SQL output pairs:

```
"input": "What's the description of the series code SM.POP.TOTL for Aruba?
(Hints: Aruba is the name of the country where ShortName = 'Aruba')"

"output": "SELECT T2.Description FROM Country AS T1 INNER JOIN CountryNotes AS T2 ON T1.CountryCode = T2.Countrycode WHERE T1.ShortName = 'Aruba' AND T2.Seriescode = 'SM.POP.TOTL'"
```

You should generate examples that examine and showcase different aspects and relation-ships of the following table schemas, described in "Table creation statements". Understand the database tables and their relationships. Understand the columns and their types and meanings to construct intresting examples.
Generate a mixture of SQL examples that include:

- some simple SQL query examples without JOIN
- some SQL query examples with aggregates, like COUNT
- some simple SQL query examples with JOIN
- some complex SQL query examples with nested JOIN

**************************
###Table creation statements###

{TARGET_DATABASE_SCHEMA}
**************************
Generate total of {k} examples. Only outputs the examples (question input and SQL output pairs), and each example can be separated by a new line.

Figure 26: Synthetic example generation prompt used for common SQL features examples genera-tion . TARGET_DATABASE_SCHEMA contains all the tables from the target database.

You are a SQLite SQL expert. Your job is to create a set of examples, where each example consists of a question and a SQL query to fetch the data for it.
You should generate examples that examine and showcase different aspects and relationships of the following table schemas. Understand the database tables and their relationships. Understand the columns and their types and meanings to construct intresting examples.
I will also show you multiple examples generated for the other database and its table schemas, so you can see what kind of examples can be generated for a given database.

*************************

###Examples from other database### The following is the table schemas and column examples for other database:
The database ({TRAIN_DATABASE_NAME}) structure is defined by the following table schemas (comments after '–' provide additional column descriptions).

{TRAIN_DATABASE_SCHEMA}

————————————

The folloiwing are the examples generated for the above database schemas:
Example 1) "input": "Among the countries in the group of Heavily Indebted Poor Countries, how many of them are under the lending category of the International Development Associations?
(Hints: group of Heavily Indebted Poor Countries is OtherGroups = 'HIPC'; International Development Associations refers to lendingcategory = 'IDA')"

"output": "SELECT COUNT(CountryCode) FROM Country WHERE LendingCategory = 'IDA' AND OtherGroups = 'HIPC'"

...

Example 10) "input": "What is the description of the footnote on the series code AG.LND.FRST.K2 in 1990 for Aruba?
(Hints: Year = 1990; Aruba is the name of country where ShortName = 'Aruba')"

"output": "SELECT T2.Description FROM Country AS T1 INNER JOIN FootNotes AS T2 ON T1.CountryCode = T2.Countrycode WHERE T1.ShortName = 'Aruba' AND T2.Seriescode = 'AG.LND.FRST.K2' AND T2.Year = 'YR1990'"

*************************
Now similarly, generate examples (question input and SQL output pairs) for the table schemas defined below, in "Table creation statements".

*************************
###Table creation statements###
TARGET_DATABASE_SCHEMA

*************************
Only outputs the examples (question input and SQL output pairs), and each example can be separated by a new line.

Figure 27: Synthetic example generation prompt. This is use TARGET_DATABASE_SCHEMA filtered with column selection result, and the model is asked to generate simple examples similar to the ones taken from the training dataset (separate from the test or dev dataset).

