# OpenReview forum: "CHASE-SQL: Multi-Path Reasoning and Preference Optimized Candidate Selection in Text-to-SQL"
_ICLR.cc/2025/Conference — ICLR 2025 Poster_

### Official Review · Reviewer_qs8r · 2024-10-20

**Soundness:** 3
**Presentation:** 3
**Contribution:** 2
**Rating:** 5
**Confidence:** 5

**Summary:**

This paper introduces Chase-SQL, which combines SQL queries generated by various LLM strategies to enhance query quality for natural language questions. It utilizes diverse approaches, including divide-and-conquer, few-shot learning with demonstration synthesis, and self-debugging. Leveraging Gemini Pro 1.5, Chase-SQL sets a new state-of-the-art performance on two prominent benchmarks, BIRD and Spider.

**Strengths:**

The main strength of the paper is its robust framework, which effectively integrates multiple LLM strategies to optimize SQL query generation, leading to state-of-the-art results. This framework not only demonstrates the versatility of techniques such as divide-and-conquer and self-debugging but also highlights the potential for real-world applications in natural language processing. It achieves superior performance on well-established benchmarks like BIRD and Spider.

**Weaknesses:**

The main weaknesses of the paper stem from the limited novelty of the framework and its individual techniques. A recent comprehensive survey on NL2SQL, available at https://arxiv.org/pdf/2408.05109, addresses many of the technical innovations discussed here, encompassing a wide range of recent studies that leverage LLMs. This suggests that the contributions may not be as groundbreaking as implied, as they largely reiterate established concepts in the field.

First, let’s discuss the individual algorithms.

W1. Divide-and-Conquer: This concept has been widely utilized in various NL2SQL studies, including DTS-SQL, TKK, and DEA-SQL, as illustrated in Figure 6 of the NL2SQL survey under the "Decomposition" branch. The papers of all the referred methods can be found in the survey. In order to highlight the technical contributions, it would be helpful to clarify how the proposed divide-and-conquer approach differs from or improves upon these existing methods.

W2. Chain-of-Thought: The Chain-of-Thought (CoT) approach has also been extensively applied in NL2SQL research, with examples such as CHESS, ACT-SQL, and COE-SQL, which can be found in Figure 6 of the NL2SQL survey under the "Chain-of-Thought" branch.

W3. Instance-Aware Synthetic Example Generation: The authors assert that they introduce a "unique instance-aware synthetic example generation." However, the use of few-shot examples has already been explored in DIN-SQL and CodeS. Additionally, rather than synthesizing examples, it is often more straightforward to select existing examples from the training data. This raises the question: what advantages does synthesis offer over selection? Please provide empirical evidence or theoretical justification for why the proposed synthetic example generation approach outperforms or differs from simply selecting examples from training data.

- DIN-SQL: https://arxiv.org/abs/2304.11015
- CodeS: https://arxiv.org/abs/2402.16347

Next, let’s explore weaknesses from other aspects.

W4. Method Ensemble: The paper claims that ensembling multiple generated SQL paths is beneficial. However, it raises the question of why not include even more methods by incorporating readily available off-the-shelf solutions.

W5. Method Ensemble vs. Module Combination: In the reference titled "The Dawn of Natural Language to SQL: Are We Fully Ready?", a method for NL2SQL automated architecture search is discussed, which explores a predefined design space of NL2SQL solutions. This prompts a consideration of the pros and cons of coarse-grained method ensembles compared to fine-grained module ensembles.

W6. Self-Consistency and Self-Reflection Methods: The authors propose self-consistency and self-reflection methods, yet both approaches have been extensively studied in the context of NL2SQL. Similarly, value retrieval and candidate generation techniques have been addressed in previous research, as noted in Table I of the aforementioned survey.

W7. Algorithm 3 Execution Comparison: In Algorithm 3, the need to "compare both (ci, cj) and (cj, ci)" raises questions. Why is it necessary to evaluate execution results in both directions? Please provide a specific example or explanation of how comparing in both directions impacts the results or addresses potential biases in the selection process.

W8. Experiments: The experimental results lack clarity regarding whether they stem from the capabilities of Gemini or the proposed framework. (1) If the benefits arise from the framework, the experiments should involve substituting Gemini with various LLMs to assess whether the framework enhances performance across different models. (2) The impact of ensembling different methods remains uncertain. (3) It is unclear how the proposed techniques individually compare to existing methods in areas such as value retrieval, candidate generation, and query fixing. Hence, the following experiments are important to enhance the experimental section:

- Conduct ablation studies comparing their framework with different LLMs.
- Provide a detailed analysis of the impact of ensembling different methods.
- Include comparisons of individual components (value retrieval, candidate generation, query fixing) against existing state-of-the-art methods for each task.

**Questions:**

Q1: The paper's novelty requires clarification, particularly in light of the concerns raised regarding the originality of its individual algorithms (see weaknesses W1, W2, and W3).

Q2: The technical contributions and key innovations compared to existing studies need to be clearly justified (refer to weaknesses W4 through W6).

Q3: The experimental design needs to be improved to more effectively verify the authors' claims (see weakness W8).

---

> ### Author Response · Authors · 2024-11-20
> **Authors' response to Reviewer (Number one)**
>
> We sincerely thank the reviewer for their insightful comments and valuable suggestions. With your valuable inputs, the quality of our paper can be improved significantly.
>
> **Novelty Concerns**:  In response, we would like to further clarify the unique contributions of our CHASE-SQL method. Our key contributions lie in the multipath candidate generation and the pairwise selector model framework. The proposed pairwise selector model, along with its selection algorithm detailed in Algorithm 3 outperforms both the well-established self-consistency method and the LLM-as-judge approach, as demonstrated in Table 7. Additionally, we introduce three novel query generators capable of producing a diverse set of candidate queries, achieving a notable upper bound accuracy of 83%. There are different designs of CoT, and our two CoT-based methods outperform prior CoT approaches for Text-to-SQL; in particular, our divide-and-conquer prompting method extends to arbitrarily complex questions by recursively breaking them into simpler sub-problems. Furthermore, we propose a novel online synthetic example generation method that dynamically generates examples based on the input question during inference. This innovative approach outperforms previous retrieval-based methods, further advancing the field.
>
> Below and in our next comments we will provide detailed responses to the mentioned weaknesses:
>
> > W1. Divide-and-Conquer: This concept has been widely utilized in various NL2SQL studies ...
>
> **Novelty in Divide and Conquer prompt**:  We appreciate reviewer bringing other "decomposition methods”.  We first wish to differentiate our divide-and-conquer chain-of-thought prompting from the approaches mentioned by the reviewer, such as DTS-SQL and DEA-SQL,  which are different ``decomposition’’ approaches than ours. These methods, as the reviewer noted, rely on a decomposition approach in handling the Text-to-SQL task **by breaking it into multiple stages**—typically schema linking, classification, and SQL generation—based on the intuition that LLMs may struggle with large amounts of information in a single prompt. While effective in some aspects, all pipeline-based approaches introduce challenges related to error propagation. Specifically, any error in an early stage, such as schema linking, can cascade, potentially impacting all subsequent stages. Recent studies, such as "The Death of Schema Linking? Text-to-SQL in the Age of Well-Reasoned Language Models [1]," highlight how such pipeline-based approaches may diminish performance with advanced LLMs like Gemini and GPT-4, which can now handle complex reasoning tasks with larger token capacities. Our method, in contrast, does not segment the task across multiple steps or require intermediary schema-linking phases. Instead, we incorporate all necessary context directly into the SQL generation phase and employ divide-and-conquer logic within a single LLM prompting call. This chain-of-thought prompting minimizes error propagation by ensuring a holistic approach where **decomposition happens solely at the SQL generation step**, improving overall reasoning performance, as evidenced in Table 4 of our results.
>
> [1] Maamari, K., Abubaker, F., Jaroslawicz, D., & Mhedhbi, A. (2024). The Death of Schema Linking? Text-to-SQL in the Age of Well-Reasoned Language Models.
>
> Furthermore, among other approaches that decompose within the SQL generation phase, such as TKK, our method diverges in that we do not rely on fine-tuning. While TKK performs decomposition via multi-task learning for each SQL clause, it still generates only a SQL query, without chain-of-thought decomposition. Moreover, fine-tuning techniques, although beneficial in certain contexts, can reduce diversity in output, limiting the effectiveness of ensemble and selection strategies such as ours. Our experiments, where we fine-tuned the Gemini-1.5-pro model and compared its Pass@20 performance with our combined generator approach, demonstrate the advantages of our model’s inherent diversity and reinforce the effectiveness of our method, which is demonstrated below:
>
> **Fine-tuning vs CHASE-SQL prompts Pass@20**: https://anonymous.4open.science/r/CHASE-SQL-REBUTTAL-7EF5/finetuning_vs_chase.png
>
> Finally, below we include the detailed comparison of our work with the methods mentioned by the reviewer using execution accuracy:
>
> | Method         | SPIDER (%) | BIRD (%)       |
> |----------------|------------|----------------|
> | DEA-SQL        | 87.1         | 52.35          |
> | DTS-SQL        | 84.4         | 55.8           |
> | TKK            | 80.3         | Not available  |
> | ChaseSQL (ours)| 87.6       | 73             |

---

> > ### Comment · Reviewer_qs8r · 2024-11-21
> > **Further Questions about Number One**
> >
> > Novelty in Multipath Candidate Generation: While generating multiple SQL queries from multiple paths is a reasonable and practical approach, its novelty is questionable. As highlighted in the survey paper (page 4, C3: Multiple Possible SQL Queries), this concept has already been well discussed in the literature.
> >
> > On the Pairwise Selector Model: The pairwise selector model is an interesting and worthwhile attempt. However, intuitively, an alternative strategy—such as combining the most effective components of different SQL queries rather than selecting a single query—might yield better results. Have you considered exploring this direction?
> >
> > Results Comparison: For context, the current #1 on the Spider leaderboard achieves 91.2%, and the #1 on the BIRD leaderboard achieves 74.12%. These results set a high bar for performance evaluation.
> >
> > On Novelty and Longevity: My key question regarding the novelty of the contributions is whether the proposed methods have the potential to endure as robust solutions, or if they are merely novel but short-lived innovations.

---

> > > ### Author Response · Authors · 2024-11-26
> > > **Authors' response to reviewers' questions**
> > >
> > > > Novelty in Multipath Candidate Generation
> > >
> > > Thank you once again for your valuable feedback and for highlighting the excellent work presented in the survey paper. As you pointed out, generating multiple SQL queries has been explored in the domain, primarily by increasing temperature or shuffling columns and tables in the database schema, as discussed in the CHESS and MAC-SQL papers. However, as we demonstrated in our experiments, simply applying the same chain-of-thought prompting (e.g., our DC CoT) while increasing temperature or shuffling the database schema results in an upper-bound performance of approximately 78%. To achieve higher diversity while maintaining high-quality candidates, integrating multiple reasoning paths is essential. As we suggested and demonstrated in our paper, this approach allows the upper bound of our generators to reach **83%**, which is significantly higher than 78%. While the concept might have been considered, multi-path CoT for text-to-SQL has not previously been implemented to achieve such an accuracy boost.
> > >
> > > > On the Pairwise Selector Model
> > >
> > > Thank you for your insightful question! Initially, we experimented with training a refiner model designed to take multiple candidate queries as input and produce either one of the original candidates or a correct query constructed by combining the candidates. While this approach has the potential to surpass the performance of pass@K, our experiments showed that it did not outperform the selector method proposed in our work. We speculate that this is primarily because selecting the best query from a set of candidates is a much simpler task than refining queries to construct a correct one and hence LLMs might yield higher accuracy on this task in robust ways. However, we believe that training models with enhanced reasoning capabilities, such as O1-style reasoning,  or advanced agentic workflows could significantly improve query refinement. This remains a promising avenue for future work, and we aim to explore it further.
> > >
> > > > Results Comparison
> > >
> > > Thank you for raising your concern about the current #1 position on the BIRD leaderboard. To address this, we trained a new selection model using multi-task learning and successfully regained the top position on the BIRD test set, reestablishing our approach as the SOTA method. This result demonstrates the robustness of our proposed method, which allows for further improvements with small modifications. To enhance the selection mechanism, we integrated the selection dataset with a refinement task, which involves transforming incorrect queries into correct ones. This combination formed the basis for training a single selection model. Our proposed multi-task learning dataset includes the following two tasks:
> > >  1. Selection Task Given a user question \\( Q_u \\), a common database schema \\( D_{ij} \\), and two candidate SQL queries \\( C_i \\) and \\( C_j \\), where one of the candidates is correct and the other is incorrect, we aim to select the correct query:
> > > \\[ C_{\\text{correct}} = F_{\\text{sel}}(D_{ij}, C_i, C_j, Q_u) \\]
> > >  2. Refinement Task In this task, given a user question \\( Q_u \\), a database schema \\( D \\), and an incorrect candidate query \\( C_{\\text{in}} \\), we aim to generate the correct SQL query \\( C_{\\text{correct}} \\):
> > >  \\[ C_{\\text{correct}} = F_{\\text{ref}}(D, C_{\\text{in}}, Q_u) \\]
> > > The refinement task helps the model become more attuned to subtle differences between incorrect and correct queries, enabling it to make precise selections at inference time. Our updated performance on the BIRD test set is **74.79**, reaffirming our approach as the state-of-the-art on this challenging benchmark.

---

> ### Author Response · Authors · 2024-11-20
> **Authors' response to Reviewer (Number two)**
>
> > W2. Chain-of-Thought: The Chain-of-Thought (CoT) approach has also been extensively applied
>
> **Novelty in DC and QP CoT**: Thank you to the reviewer for highlighting relevant works in Chain-of-Thought (CoT) approaches. While our proposed CoT methods are conceptually similar to previous works, they differ significantly in the design of the reasoning process, as the reasoning steps and implementation of the 'chain of thought' in our approach are fundamentally distinct: DC prompt reasons the task of Text-to-SQL as solving sub-SQL and writing pseudoSQLs recursively; QP prompt reasons the task “Text-to-SQL” as the way database engine execute the SQL; all of these reasoning methods are significantly different compared with CoE-SQL, DIN-SQL, and ACT-SQL. We note that CoE-SQL is inherently designed for multi-turn Text-to-SQL, where it depends on iterative unit edits across dialogue turns, which differs from our single-turn task on datasets like BIRD and Spider, so this approach is not considered as CoT for single-turn text-to-SQL. Similarly, CHESS uses zero-shot CoT without intermediate reasoning steps, which we have already addressed in our comparative analysis and compared with as our baseline (Table 4) and table below. Finally, ACT-SQL proposes a specific CoT prompt, which is similar to DIN-SQL, so we decided to compare our CoTs with this method to showcase our innovation. For this comparison, as ACT-SQL is only proposed for the Spider dataset, we implement their CoT for the BIRD benchmark using the same set of few-shot samples as our CoTs and provide the results in the table below. **Our approach outperformed ACT-SQL CoT**, demonstrating the effectiveness of our CoT design for Text-to-SQL. Results below demonstrate the effectiveness of our proposed CoT designs in comparison to the previous works, where the detailed step-by-step decomposition resulted in roughly 2% improvement over the well-established ACT-SQL (DIN-SQL) CoT. Additionally, our online example generation approach significantly outperforms the baseline with 6% gap:
>
> | Method                           | Execution Accuracy (%) |
> |----------------------------------|-------------------------|
> | Baseline (zero-shot CoT) + Gemini 1.5 pro | 57.75                  |
> | QP CoT + Gemini 1.5 pro          | 63.62                  |
> | DC CoT + Gemini 1.5 pro          | 63.92                  |
> | OS + Gemini 1.5 pro              | 67.09                  |
> | ACT-SQL + Gemini 1.5 pro         | 61.60                  |
>
> Finally, we would like to also compare our approach directly to the works mentioned by the reviewer,using execution accuracy as our metric, to showcase our improvements:
>
> | Method         | SPIDER (%) | BIRD (%)      |
> |----------------|------------|---------------|
> | ACT-SQL        | 82.9       | Not available |
> | CHESS          | 87         | 65            |
> | CoE-SQL        | Not applicable | Not applicable |
> | ChaseSQL (ours)| 87.6       | 73            |

---

> > ### Comment · Reviewer_qs8r · 2024-11-21
> > **Further Questions about Number Two**
> >
> > Chain-of-Thought (CoT) prompting is a form of prompt engineering. While stating that "our approach outperformed ACT-SQL CoT" demonstrates the results of your method, it does not justify its novelty. Your approach may indeed represent a new variation of prompt engineering, but I wouldn’t consider it "fundamentally different" from existing methods.
> >
> > It is also worth noting that the current #1 entries on both the Spider and BIRD leaderboards have already surpassed CHASE-SQL. Given this context, I am looking for truly fundamental contributions in CHASE-SQL that set it apart from other methods and have the potential for broader, lasting impact.

---

> ### Author Response · Authors · 2024-11-20
> **Authors' response to Reviewer (Number three)**
>
> > W3. Instance-Aware Synthetic Example Generation
>
> Thank you for the invaluable comments and also the references. We noticed the gap in our exposition as well as in the study, and we tried again to clarify the following regarding the novelty and effectiveness of our synthetic example generation strategy:
> In Section 3.3 > Online Synthetic Example Generation, we have clarified how our example question-SQL pairs are different from few-shot “demonstrations” proposed by DIN-SQL (please follow the citation in the text). Manually crafted few-shot demonstrations are effective in specifying the task and step-by-step process, whereas we generate/synthesize more common few-shot examples, illustrating the mapping between input and output. We have added more references, including your pointers. Our approach, unlike typical few-shot in-context learning (ICL) for Text-to-SQL, generates way more than 5 examples, and we draw the connection to the recent many-shot ICL work for other application domains (QnA. translation, summarization, etc.). While prior few-show ICL focuses on retrieval of relevant examples by question similarity or some variations of it (e.g., masking the keywords, as suggested in CodeS and others), we instead generate examples on-the-fly per incoming question. This means we bypass the error-prone selection process and pass all the examples we generate for the given question. We found this strategy and our recipe for example generation effective in terms of the final accuracy.
>
> In the original text, we tried to emphasize that we are more strictly guiding the generation process for diversity of the SQL structures and the use of schema elements. This is very different from prior works for synthetic data augmentation, where LLMs are asked to come up with feasible questions and generate SQL for them; and then LLMs are asked to fill in the blanks using relevant schema elements given a universal set of SQL templates – here the templates are very limited in complexity (1 table, unnested queries only). Instead, we explicitly look to generate SQLs and examples following guidelines detailed in our text. Diversity is the key for us to help generations. To that end, thank you for the suggestion - we have also added another ablation study comparing our examples (ICL quality) vs. similar examples drawn from training dataset, which is the common techniques used for ICL works using BIRD bench (Appendix Table 9) – note that CodeS also uses training data for own ICL evaluation. Since they included a cross-domain data augmentation strategy (not evaluated for BIRD), we also try to implement the strategy (CodeS, thank you for the reference) to show how the limited complexity examples perform against us.
>
> <Appendix Table 9. Comparison study of the proposed synthetic example vs. selected training data examples vs. example synthesis technique from CodeS>: https://anonymous.4open.science/r/CHASE-SQL-REBUTTAL-7EF5/OS_figure1.png
>
> <Appendix Figure 26 to show how our generated example SQL feature distribution compares with the ground truth and other example generation strategy>: https://anonymous.4open.science/r/CHASE-SQL-REBUTTAL-7EF5/OS_figure2.png
>
> > W4. Method Ensemble
>
> **Ensemble and why not more**: We should highlight that, based on our experiments reported in Table 1, our current operating regime represents a near-optimal trade-off between cost and accuracy. Though our framework is designed in a way that users can freely add more methods. As shown in Table 1, the Pass@21 performance with our three generators reaches approximately **83% accuracy**, significantly surpassing the current SOTA performance. This highlights that our generators already deliver high performance, and the primary challenge lies on the selection side. Here, our innovative pairwise selector model achieves better performance than the self-consistency approach. Additionally, as detailed in our error analysis in the Appendix and similar analyses in works like CodeS and CHESS, there is a notable presence of ambiguous questions and incorrect golden SQL queries, underscoring that 83% accuracy is already very high. Thus, adding more generators of some variations would add marginal or no improvements, unless they perform differently from our generators on the ambiguous questions. Additionally, adding more candidate generators will induce more cost not only in the generation step but also in the selection step, which makes it less desirable given the diminishing return.

---

> ### Author Response · Authors · 2024-11-20
> **Authors' response to Reviewer (Number four)**
>
> > W5. Method Ensemble vs. Module Combination:
>
> **Ensemble components vs end2end methods (ours)**: Thank you for your comment and the reference. While studying single pipeline components is valuable—given that text-to-SQL pipelines typically involve multiple components or steps, and different combinations for each step can yield varying results, as demonstrated in NL2SQL360—our focus lies elsewhere. Specifically, we concentrate on the optimization of a “coarse-grained methods ensemble” rather than single pipeline optimization (referred to as a ‘method’ by the reviewer and ‘generator’ in our text). Our framework emphasizes building a diverse candidate pool by leveraging outputs from various generations, with pipeline optimization (or component ensemble) considered orthogonal to our approach. If a generator contributes unique answer candidates, it can be integrated into our framework to further enhance performance. As explained in W4, we selected three generators to demonstrate the effectiveness of our approach, showing that our “coarse-grained methods ensemble” can outperform the best results from other single generator baselines. Compared to NL2SQL360, our design search space is significantly larger, supporting multiple prompts or pre/post-processing techniques simultaneously rather than selecting a single configuration for each component. This broader flexibility enables our framework to achieve superior performance, with a 73% score on BIRD-bench compared to NL2SQL360's 58.5%, even though NL2SQL360 used both GPT-4 and fine-tuned PLMs, while we used Gemini-1.5. In summary, our work proposes a novel framework that improves Text-to-SQL accuracy,  while leaving room for future pipeline optimizations to enhance our framework further when incorporated as additional generators.
>
> > W6. Self-Consistency and Self-Reflection Methods
>
> **Self-Consistency and Self-Reflection Methods are not our novel contribution**:   In our paper, we used self-consistency as a robust **“baseline”** to compare with our novel pairwise selector model; self-consistency was never presented as our contribution, and we will mention this in the updated paper to avoid confusion. Additionally, for value retrieval, we explicitly referenced in lines 144 and 145 that our approach aligns with that proposed in the CHESS paper. Similarly, regarding the self-correction module, as noted in line 250, we stated that self-reflection is a commonly used method for enhancing Text-to-SQL approaches. Our key contributions lie in the multipath candidate generation and the pairwise selector model, which not only outperforms the well-established self-consistency but also surpasses the LLM-as-judge approach, as shown in Table 7. Additionally, self-reflection, “fixer”, is one of working components of many text-to-SQL methods, is not our key novelty. We hope this clarification addresses any confusion.
>
> > W7. Algorithm 3 Execution Comparison
>
> Algorithm 3 Execution Comparison: Large language models can exhibit order bias when selecting between candidates, as noted in prior works such as "Unveiling Selection Biases: Exploring Order and Token Sensitivity in Large Language Models"[2]. For example, when an LLM is presented with (ci, cj), it may choose ci, but when the order is reversed to (cj, ci), it may favor cj, reflecting a bias towards the first candidate. If we only consider one directional comparison, either (ci, cj) or (cj, ci), this bias can skew the selection process, favoring one query simply due to its position and thus affecting our scoring mechanism. By considering both orderings, we assign scores to each candidate more equitably, effectively reducing order bias. To illustrate this impact, we conducted a one-way comparison, and the final execution accuracy on the dev set dropped from **73.01% to 71.12%**, underscoring the importance of mitigating order bias. We also included this experiment in the paper to help the readers understand the importance of two-way comparison.
>
> [2]: Wei, S.-L., Wu, C.-K., Huang, H.-H., & Chen, H.-H. (2024). Unveiling Selection Biases: Exploring Order and Token Sensitivity in Large Language Models.

---

> ### Author Response · Authors · 2024-11-20
> **Authors' response to Reviewer (Number five)**
>
> > W8. Experiments
>
> In order to address your concern about other Large language models, we have implemented the CHASE-SQL method using the Mistral-large model. For the pairwise selector model we trained a Qwen2.5-coder 7B model. The results highlight that our pairwise query selection approach significantly improves performance, achieving **SOTA results with open-source models** on the BIRD benchmark, as detailed below: We have included the open source table in the updated draft.
>
> | Method                                   | Execution Accuracy (%) |
> |------------------------------------------|-------------------------|
> | Basic prompt                             | 54.88                  |
> | Basic prompt + fixer                     | 60.03                  |
> | Divide and conquer prompt                | 58.99                  |
> | Divide and conquer prompt + fixer        | 63.75                  |
> | Query plan prompt                        | 59.64                  |
> | Query plan prompt + fixer                | 62.64                  |
> | Online synthetic                         | 56.32                  |
> | Online synthetic + fixer                 | 61.47                  |
> | CHASE-SQL + Self-consistency             | 67.60                  |
> | CHASE-SQL + Gemini flash Selector        | 68.90                  |
> | CHASE-SQL + Qwen-2.5-Coder 7B selector   | 70.33                  |
>
> We hope our detailed response addresses your concern. We would greatly appreciate it if you could update the scores accordingly
>
> > Questions:
>
> * We have already included the ablation studies with open-source models as provided above.
> * Regarding the ensembling of different methods, we have included a detailed ablation study in Table 7, demonstrating how performance changes when each generator is removed from our proposed method.
> * As stated in our response to w6, self-correction and database value retrieval are not contributions of our work, so comparisons with previous works on these aspects are not relevant. For candidate generators, as highlighted in the table above (our first comment), we compared our novel CoT approaches with two baselines, demonstrating the significance of each CoT. Please note that our key contributions are the pairwise selection and multi-path reasoning generation methods.

---

> ### Comment · Reviewer_qs8r · 2024-11-21
> **Further Questions about Number Five**
>
> Thank you for addressing my previous feedback and providing the new experiments. Based on your response, I’d like to raise my score to 5.

---

### Official Review · Reviewer_s8pW · 2024-10-30

**Soundness:** 4
**Presentation:** 4
**Contribution:** 3
**Rating:** 8
**Confidence:** 5

**Summary:**

The paper introduces CHASE-SQL, a state-of-the-art Text-to-SQL approach that achieved top performance on the BIRD benchmark at the time of submission. It employs three prompting strategies — Divide-and-Conquer, Query Plan, and Few-Shot prompting with synthetic example generation — to first generate a diverse set of SQL candidates. These candidates are then evaluated by a selector, which is fine-tuned on BIRD to select the correct SQL from two SQLs. The selector does pair-wise comparison over the pool of candidates and outputs the one having the highest score.

**Strengths:**

1. The paper is well-written and provide a lot of details and insights for readers to learn about. The evaluation is solid with comprehensive comparison, lower/upper bound analysis and ablation studies. The design of all components within CHASE-SQL are well-justified.

2. Although the pipeline of generating a candidate pool and then selecting from it is not new, CHASE-SQL introduces novel strategies for candidate generation and fine-tunes the selector in a unique way, focusing on simple tasks, i.e., pairwise selection, rather than relying on reranking or selecting from a large pool. I particularly appreciate the approach used to construct the few-shot examples. Instead of tailoring examples to specific SQL types, it includes examples for both the full database and the specific database relevant to the question. Moreover, it aims to cover a broad range of SQL features rather than only complex examples, which significantly reduces the risk of the model overfitting to the provided examples.

3. By using examples generated on BIRD and fine-tuning the selector solely on BIRD, CHASE-SQL also achieves competitive performance on the Spider benchmark, demonstrating its generalizability.

**Weaknesses:**

The main concern is the cost and latency of CHASE-SQL. Assuming each generator produces 7 candidates, this results in 21 LLM calls. The prompt length for each call is also substantial, especially with the few-shot prompting strategy, which includes examples for both the full database and the specific database, aiming to cover a wide range of SQL features. For the selector, CHASE-SQL employs a pairwise comparison strategy, leading to O(n^2) LLM calls, where n is the total number of generated candidates. It would be helpful if the authors reported the total number of tokens processed by CHASE-SQL to generate SQL for a user query and provided the end-to-end latency. Given these factors, I am uncertain whether CHASE-SQL can achieve interactive Text-to-SQL.

**Questions:**

1. How many examples are included in the few-shot prompting (online synthetic example generation)?
2. What is the total number of tokens processed if I use CHASE-SQL to generate the SQL for a single question?
3, What is the end-to-end latency of CHASE-SQL?

---

> ### Author Response · Authors · 2024-11-20
> **Authors' response to reviewer**
>
> We sincerely thank the reviewer for their valuable comments and suggestions, we really appreciate it.
>
> > The main concern is the cost and latency of CHASE-SQL ...
>
> Regarding the cost analysis of our approach, we compared it to the CHESS method, a former SOTA method on text-to-SQL benchmarks, and demonstrated that our method consumes fewer tokens, as shown below. Additionally, the cost of generating queries with human annotation remains significantly higher than the cost achieved with our proposed approach. Moreover, we should consider that LLM cost has been reduced significantly during the past years, so we speculate that approaches with higher inference-time computation like ours will be adopted more. Additionally, we provided a detailed latency analysis of our proposed methodology on all of the BIRD development set databases. Based on this analysis, we identified that for the databases where query fixing was required more than the other databases the latency is higher as the query fixing step is a sequential process. To address concerns about token usage and latency, we have included a detailed analysis below:
>
> **Detailed Cost Analysis of CHASE-SQL**:  https://anonymous.4open.science/r/CHASE-SQL-REBUTTAL-7EF5/token_usage.png
>
> **Detailed Latency Analysis of CHASE-SQL**: https://anonymous.4open.science/r/CHASE-SQL-REBUTTAL-7EF5/Latency_Analysis.png
>
> > O(n^2) LLM calls, where n is the total number of generated candidates ...
>
> For SQL generation, the 21 candidates are iid samples, allowing for parallel generation with negligible overhead compared to single-candidate generation. While the pairwise comparison step could have a worst-case complexity of O(n^2), as noted in line 3 of Algorithm 3, comparisons are skipped for queries with identical execution results, significantly reducing computational time since most candidate queries yield the same results.
>
> We hope our response addresses your concern. We would greatly appreciate it if you could update the scores accordingly.
>
> > Questions:
>
> * **How many examples**: For each of the Query plan CoT and the divide and conquer CoT we included 8 examples in the prompt. For OS, we generated a total of 75 examples per user question. We discuss the choice in Appendix A.12 and Tables 8 and 9.
>
> * **What is the total number of tokens processed**: Figure above provides the average number of tokens used for our approach on all of the BIRD databases.

---

> ### Comment · Reviewer_s8pW · 2024-11-21
> **Further question about the cost analysis**
>
> I appreciate the authors' effort in adding new results to the paper. However, I have concerns regarding the token cost analysis presented for the Chase-SQL Generator.
>
> According to the figure, for the financial database, the total number of input tokens for the Chase-SQL Generator amounts to approximately 0.16 million across 106 questions. This calculates to an average of about 1,510 tokens per question. This number is significantly lower than what I expected.
>
> To better understand this, I analyzed the generator input tokens, which comprise the lengths of the Divide-and-Conquer Prompt, Query Plan Prompt, and Few-Shot Prompt. Specifically, I generated a database description string for the financial database, which includes basic schema information, simple column descriptions, and a few value examples, totaling 1,204 tokens.
>
> Upon integrating this database description into the provided prompt from your paper, I calculated the prompt token length for a single question as follows:
>
> Divide-and-Conquer Prompt: 2,055 tokens
> Query Plan Prompt: 1,814 tokens
> Few Shots Prompt: I compute the token length for one example in the paper, which has 126 tokens. The author says they generate 75 examples for each user question. Thus, this prompt is at least 9450 tokens.
>
> Based on these components, the total token count per question exceeds 13319 tokens (much larger than the figure shows). Could you please clarify this, or have I misunderstood the token computation method described?

---

> > ### Author Response · Authors · 2024-11-21
> > **Authors' response to the reviewer**
> >
> > Thank you for your valuable feedback on the token cost analysis.
> >
> > To clarify, the token analysis provided represents an average per question, not a cumulative total across all questions. For the financial database, **each question** requires approximately 0.16 million tokens solely for the input of the generators, not for the entire set of questions in the database.
> >
> > I hope this explanation helps clear up any confusion regarding token usage. Please let us know if you have further questions.

---

### Official Review · Reviewer_2iLv · 2024-11-04

**Soundness:** 3
**Presentation:** 4
**Contribution:** 3
**Rating:** 6
**Confidence:** 4

**Summary:**

This paper introduces the CHASE-SQL framework, a novel approach for improving text-to-SQL task with LLMs. The framework proposes multi-path reasoning techniques that decompose complex queries and optimize candidate generation for SQL, which involves three main strategies: a divide-and-conquer approach for breaking down queries, a CoT reasoning based on query execution plans, and an instance-aware synthetic example generation to enhance LLM performance. To select the best SQL candidate, a fine-tuned selection agent ranks generated queries through pairwise comparisons, achieving high accuracy. Extensive experiments have validated the effectiveness of the proposed framework. CHASE-SQL demonstrates SOTA performance on the well-recognized BIRD benchmark, achieving an execution accuracy of 73%, making it the top place on the leaderboard.

**Strengths:**

1. The performance of the proposed CHASE-SQL framework is highly promising. It achieves SOTA accuracy on the well-recognized BIRD benchmark, outperforming both published and undisclosed methods, thereby demonstrating its effectiveness in complex text-to-SQL tasks.

2. The experiment of the paper is sufficient, and the error analysis section is detailed and highly informative. The authors provide comprehensive analyses, including performance across different databases, error analysis, and selection agent errors, offering clear insights into the model's strengths and areas for improvement. Notably, the error analysis section is outstanding and is garnering increasing attention within the text-to-SQL community, as it helps readers understand not only how the framework works but also why it works.

3. The overall framework is novel, and the paper is well-organized and easy to understand. The structure effectively presents the methodology, experiments, and results, making it accessible for readers to comprehend the contributions and significance of the proposed approach.

**Weaknesses:**

1. The paper does not include the cost of the proposed framework. Prompting proprietary LLMs for SQL generation has become a mainstream approach in text-to-SQL research. When the performance of various methods shows no significant differences, the method with lower API costs is typically preferred. Previous work has focused more on models released by OpenAI (e.g., ChatGPT, GPT-4) [1]. Since the Gemini series is also an outstanding proprietary LLM, this paper presents a good opportunity to introduce the series to the community by comparing its performance and API costs.

2. The model ablation study is lacking in the paper, and the selection of open-source models has not been discussed. The paper would benefit from verifying the proposed framework on a broader range of models, including GPT series models (e.g., ChatGPT, GPT-4) and open-source models (e.g., LLaMA-3.1, Qwen-2.5). Recently, well-designed frameworks for open-source models have achieved promising progress [2][3], demonstrating particular effectiveness for local deployment and real-world applications. Therefore, experiments on open-source models in this paper could further advance this development.

3. The description of the Query Fixer module is relatively brief. The authors could consider adding a detailed algorithm for correcting incorrect SQL in Section 3.4. Additionally, I suggest including a separate limitations section to discuss the potential challenges for application, such as the framework's complexity, generalization and extension capabilities [4], and the potential limitations in handling ambiguous questions [5][6].

[1] BIRD Leaderboard. https://bird-bench.github.io/

[2] Haoyang Li, et al. "CodeS: Towards Building Open-source Language Models for Text-to-SQL" In Proceedings of SIGMOD, 2024.

[3] Mohammadreza Pourreza, et al. "DTS-SQL: Decomposed Text-to-SQL with Small Large Language Models" arXiv preprint, 2024.

[4] Zijin Hong, et al. "Next-Generation Database Interfaces: A Survey of LLM-based Text-to-SQL" arXiv preprint, 2024.

[5] Yujian Gan, et al. "Towards Robustness of Text-to-SQL Models against Synonym Substitution" In Proceedings of ACL, 2021.

[6] Xiang Deng, et al. "Structure-Grounded Pretraining for Text-to-SQL" In Proceedings of NAACL, 2021.

**Questions:**

1. As a reviewer, I highly appreciate the detailed error analysis in this paper. However, I suggest that the authors include some parts of the error analysis in the main content instead of the Appendix, as it is gaining increasing attention from the text-to-SQL community.

2. As shown in Appendix A.3, why is the number of correct samples for some databases zero across various methods? Could this result be combined with the difficulty level of the corresponding questions for further analysis?

3. For the prompt used in the Query Fixer module, there are few-shot examples provided, as shown in Appendix A.7. What type of execution results do the examples represent? How does the performance of this module vary across different types of execution results (errors)? As an assumption, could including a diverse range of few-shot examples that cover as many types of execution errors (e.g., Column Not Found, Data Type Mismatch) enhance the correcting capability of the Query Fixer module?

4. The authors could consider using a different abbreviation instead of "CHASE" since there is a related work in the text-to-SQL community with a similar name [7].

[7] Jiaqi Guo, et al. "Chase: A Large-Scale and Pragmatic Chinese Dataset for Cross-Database Context-Dependent Text-to-SQL" In Proceedings of ACL, 2021.

---

> ### Author Response · Authors · 2024-11-20
> **Authors' response to reviewer**
>
> We sincerely thank the reviewer for their valuable feedback and comments on our paper.
>
> > The paper does not include the cost of the proposed framework ...
>
> As for the cost analysis, we have conducted a detailed token usage (noting that dollar costs are proportional to that) estimation comparing our method with the CHESS method, demonstrating that our approach uses fewer input tokens than previous SOTA approaches, as provided below: We have also included the figure in appendix of updated draft.
>
> **Link to the Cost Analysis experiment**: https://anonymous.4open.science/r/CHASE-SQL-REBUTTAL-7EF5/token_usage.png
>
> >  selection of open-source models has not been discussed ...
>
> As requested by the reviewer, to further demonstrate the effectiveness of our approach with open-source large language models, we have conducted the performance analysis of CHASE-SQL using the **Mistral-large model**. For the pairwise selector model we trained a **Qwen-2.5-coder 7B** model and also compared it with Gemini-1.5-flash model. The results highlight that our pairwise query selection approach significantly improves performance compared to well-established self-consistency, achieving **SOTA performance with open-source models** on the BIRD benchmark, as detailed below: We have included the open source table in the updated draft.
>
> | Method                                   | Execution Accuracy (%) |
> |------------------------------------------|-------------------------|
> | Basic prompt                             | 54.88                  |
> | Basic prompt + fixer                     | 60.03                  |
> | Divide and conquer prompt                | 58.99                  |
> | Divide and conquer prompt + fixer        | 63.75                  |
> | Query plan prompt                        | 59.64                  |
> | Query plan prompt + fixer                | 62.64                  |
> | Online synthetic                         | 56.32                  |
> | Online synthetic + fixer                 | 61.47                  |
> | CHASE-SQL + Self-consistency             | 67.60                  |
> | CHASE-SQL + Gemini flash Selector        | 68.90                  |
> | CHASE-SQL + Qwen-2.5-Coder 7B selector   | 70.33                  |
>
> > The description of the Query Fixer module is relatively brief ...
>
> Thank you for your insightful suggestion. We have updated the paper by adding a limitations section and details about the query-fixing algorithm. Regarding the limitations of our work, as you noted, most current text-to-SQL systems assume that user questions are inherently answerable, and we have now included this as a limitation in the paper. Furthermore, the current framework has several limitations that are open avenues for future work. Adapting to additional SQL dialects poses challenges due to their unique syntactic and semantic variations, necessitating automated adaptation techniques for the proposed modules. Reducing latency is another critical area, which could be achieved through optimized prompt engineering and the use of smaller models via distillation. Finally, integrating agentic design into foundation model development could mitigate train-test mismatches by enabling models to actively query and refine their understanding during training, thereby enhancing robustness and alignment with real-world applications.
>
> We hope our response adequately addresses your concerns. If all your concerns have been resolved, we would greatly appreciate it if you could update the scores accordingly.
>
> > Questions:
>
> * **error analysis**: Sure, we will bring a portion of the error analysis section in the main paper for the final version of our paper.
> * **Appendix A.3**: Thank you for bringing this to our attention. We believe this figure requires additional explanation to avoid confusion. In this section, we present the number of samples across different databases where only one of the candidate generators produces a correct result, meaning the other two generators fail to provide a correct answer. A value of zero for any generator in this figure indicates that whenever that generator produces a correct result, the other two generators also manage to generate at least one correct answer. We updated this section in the Appendix section accordingly.
> * **prompt used in the Query Fixer**: As you suggested, including examples of each error type can certainly enhance the performance of the fixer module. In our approach, since the majority of the error cases we observed were related to "column not found" errors, we focused on including most of the few-shot samples from this category.
> * **"CHASE" abbreviation**: Thank you for your suggestion regarding the name and your consideration of potential confusion. Our method has already been submitted to public leaderboards and gained visibility under its current name, so changing it at this stage might be less ideal. However, we will carefully evaluate your feedback and give it further thoughtful consideration.

---

### Official Review · Reviewer_xZFJ · 2024-11-04

**Soundness:** 3
**Presentation:** 3
**Contribution:** 3
**Rating:** 6
**Confidence:** 5

**Summary:**

The article discusses the formation of a pipeline for text2sql, which includes the following main parts: combining multiple methods (including DC-CoT, Query Plan-CoT and OS-CoT) to generate a set of candidate sql, with each sql query running through a fixer to fix the query so that it can be run, and finally, comparing each pair of queries and ranking to choose the most suitable query. The pipeline is implemented on close-source LLMs and is carefully promoted and evaluated to come up with a good method when testing multi-agent LLM for the text2sql task. The method reaches an execution accuracy of 73.01% and 73.0% on the development set and test set of the challenging BIRD Text-to SQL dataset which outperforms all of the published and undisclosed methods on this benchmark by a large margin.
Contribution:
+ Experiment multi method (include few-shot, CoT) to generate candidate SQL query from natural language text.
+ Develop evaluator agent to score every candidate and ranking for choose the best candidate.
+ Experiment combination and ablation many method choose to the effective text2sql pipeline.
+ Adapt Multi-agent LLM approach.

**Strengths:**

+ Proposed a pipeline that provides high efficiency on text2sql data and has proven its efficiency and high results on difficult benchmarks such as Bird and Spider.
+ There is a thorough review and explanation, making it easy to combine and remove components throughout the system.
+ The paper is presented fully and in detail, with thorough explanations of mathematical formulas, complete conclusions and details in the evaluation and testing.

**Weaknesses:**

+ CoT, Few Shot or query fixer methods are quite classic in testing LLM, so they have not mentioned the highlights in generating this query.
+ Beside the method uses available closed source models such as claude, gemini pro mainly to perform prompting and connect agents together, it is not possible to verify whether it is good to fine-tune open source for the above methods. The article would be better if there was more in-depth research on fine-tuning these models, which could be a useful scientific research and could be more easily deployed in practice. Overall, I rate the article very well in terms of testing and conclusions drawn.

**Questions:**

+ How about experimenting with the above pipeline for an open-source LLM, is it possible?
+ Can you give some examples of selecting rows to match the schema and user query?
+ Why schema union is so effective in evaluating your query pairs.

---

> ### Author Response · Authors · 2024-11-20
> **Authors' response to reviewer**
>
> Thank you so much for your detailed comments that have helped us to improve our submission, we truly appreciate them. We hope the detailed answers and additional results provided below address your concerns. We kindly ask you to consider the possibility of a score adjustment
>
> >  it is not possible to verify whether it is good to fine-tune open source for the above methods ...
>
> To address your point regarding the performance of our proposed methodology on open-source models, we have included new results using the Mistral-large model in combination with the fine-tuned Qwen2.5-coder model as the pairwise selector. These results align with the findings reported in the paper (Table 4 and Table 6) for other models, demonstrating that our proposed methods—Divide-and-Conquer Prompt, Query Plan, and Online Synthetic Prompt—consistently outperform the Basic Prompt by a significant margin. Furthermore, our pairwise selection approach surpasses the well-established self-consistency method when evaluated with both the fine-tuned Gemini-1.5-flash and Qwen-2.5-coder models.
>
> We note that the reported results below are the **SOTA performance with open-source LLMs** on the BIRD benchmark. We have reported the performance by generating the candidate queries with the **Mistral large** model and included two selectors of **Gemini-flash** and **Qwen-2.5-coder** models below: We also included this open source table results in our updated paper.
>
> | Method                                   | Execution Accuracy (%) |
> |------------------------------------------|-------------------------|
> | Basic prompt                             | 54.88                  |
> | Basic prompt + fixer                     | 60.03                  |
> | Divide and conquer prompt                | 58.99                  |
> | Divide and conquer prompt + fixer        | 63.75                  |
> | Query plan prompt                        | 59.64                  |
> | Query plan prompt + fixer                | 62.64                  |
> | Online synthetic                         | 56.32                  |
> | Online synthetic + fixer                 | 61.47                  |
> | CHASE-SQL + Self-consistency             | 67.60                  |
> | CHASE-SQL + Gemini flash Selector        | 68.90                  |
> | CHASE-SQL + Qwen-2.5-Coder 7B selector   | 70.33                  |
>
> > Questions:
>
> * **Open Source model**: Please see the results above for open-source LLMs which is also included in the updated paper.
> * **Query-specific database values retrieval**: We include an example here and also updated the Appendix section to include more details. Value retrieval is an important step in our proposed pipeline as it can help to identify the relevant columns and tables to the user’s question and also provides the correct filtering for the SQL conditions. For the question: “What is the highest eligible free rate for K-12 students in the schools in Alameda County?” from the “california_schools” database, using the value retrieval, we get the following results, showing the retrieved database values from different columns:
>   * SOC Type: Preschool
>   * EIL Name: Preschool
>   * Schools: Preschool, MethodSchools, Alameda County Community, Alameda County Opportunity, Alameda High
>   * Mail Street Address: 4600 Student Lane
>   * Street Address: 4600 Student Lane
>   * Mail City: Alameda
>   * City: Alameda
>   * Grades Served: K-12
>   * Grades Offered: K-12
>   * County: Alameda
>   * Districts: Alameda Unified, Tri-County ROP
> * **Design choice of schema union in Selection Agent**: For the pairwise comparison of the candidates, we have decided to use the union schema, as columns in the database that are not used by both candidates would not affect the comparison decision. This way we also reduce the token usage and avoid any irrelevant information in the prompt.

---

### Author Response · Authors · 2024-11-20
**Authors' comment about paper update**

Dear Meta-Reviewer and Reviewer,

Thank you for your valuable comments and feedback. We have updated our paper accordingly, and the revised version is now available for review in the PDF above. All changes are highlighted in **red text** for easy identification.

The key updates are as follows:

* Adding the new experiments with totally open-source models where we achieved SOTA performance with open-source model on BIRD. Updated sections: 1) Introduction 2) Table 2 3) Table 4 4) Experiments section

* Detailed ablation studies for Online synthetic example generation method: Appendix Section A.12

* Adding Query fixer algorithm, section 2.4

* Adding more description about the novelty of Online synthetic example generation method, updated section 2.3

* Adding description about the importance of both-direction comparison for selection agent: updated sections 2.5

* Limitations and future works: Appendix Section A.1

* Value retrieval Example: Appendix Section A.4

* Token usage analysis: Appendix Section A.6

---

### Meta-Review · Area_Chair_hE46 · 2024-12-20

**Metareview:**

This paper proposes the CHASE-SQL framework, which improves text-to-sql tasks using LLMs. Overall the reviews are quite positive as CHASE-SQL is novel, efficient, and has good results on benchmarks like Bird and Spider. The paper is also well written and provides plenty of detail and insights. I therefore recommend accepting the paper.

**Additional Comments On Reviewer Discussion:**

Reviewer qs8r had the most concerns on the paper in terms of novelty, but the authors made significant efforts to address them, which resulted in a score increase.

---

### Decision · Program_Chairs · 2025-01-22

Accept (Poster)